# Mitochondrial genomics and phylogeny of noctuoid moths: Implications for Macroheterocera

**Sivasankaran Kuppusamy**[1]*, **Muzafar Riyaz**[1], **Rauf Ahmad Shah**[1],
**Ignacimuthu Savarimuthu**[2], **Karuppasamy Paraman**[1]

**1** Division of Taxonomy and Biodiversity, Entomology Research Institute, Loyola Collège, Chennai, Tamil Nadu, India, **2** Xavier Research Foundation, St. Xavier's College, Palayamkottai, Tamil Nadu, India

* ganesh_swamy2005@yahoo.com, sivasankaran@loyolacollege.edu

## Abstract

The majority of the Lepidoptera species belongs to the Macroheterocera clade. The macroheteroceran superfamilies' phylogenetic relationships are still unstable. The construction of a robust phylogenetic tree and comprehensive analysis can be facilitated by an increased availability of mitochondrial genome data. In this study, the mitochondrial genomes of five species such as *Episparis tortuosalis, Pandesma quenavadi, Erebus macrops, Polydesma boarmoides* and *Xanthodes albago* from two families in the superfamily Noctuoidea were sequenced, assembled, and annotated. The mitochondrial genomes have characteristic circular double-stranded structures observed in other lepidopteran moths, including 13 protein-coding genes, 22 transfer RNAs, two ribosomal RNAs, and the control region. All PCGs typically start with ATN codons, but *nad4* and *nad4l* in *E. tortuosalis* are not starting with standard initiation codons. Phylogenetic analysis was performed using Bayesian Inference (BI) and Maximum Likelihood (ML) through PhyloSuite v1.2.3 based on amino acid sequences of 13 mitochondrial PCGs. The tree indicates close ancestry of *E. tortuosalis* with Noctuidae insects rather than with Erebidae. Major superfamilies in Macroheterocera and their phylogenetic relationships were as follows: ((Geometroidea)+ ((Lasiocampoidea+ Bombycoidea)+ (Drepanoidea)+ (Noctuoidea))))); this showed a novel relationship compared to previous analyses. This analysis significantly enhanced the Noctuoidea mitogenome database and reinforced the high-level phylogenetic relationships of macroheterocera clade.

## Introduction

Lepidoptera is the second largest order after Coleoptera; it has more than 158,570 described species belonging to 137 families among 43 superfamilies [1,2]. Numerous high-level phylogenetic clades have been defined for Lepidoptera, which includes

**Data availability statement:** The mitogenomes of five species for this study are publicly available from the following NCBI repositories: Episparis tortuosalis (https://www.ncbi.nlm.nih.gov/nuccore/MW879209); Pandesma quenavadi (https://www.ncbi.nlm.nih.gov/nuccore/MW899034); Erebus macrops (https://www.ncbi.nlm.nih.gov/nuccore/MW924115); Polydesma boarmoides (https://www.ncbi.nlm.nih.gov/nuccore/MW969649); Xanthodes albago (https://www.ncbi.nlm.nih.gov/nuccore/MW813975).

**Funding:** KS, SI received the fund Grant number: EMR/2017/000566 Department of Science and Technology (DST)-Science and Engineering Research Board (SERB) https://serb.gov.in/ Funding agency has no role in the study design, data collection and analysis, decision to publish, preparation of the manuscript.

**Competing interests:** The authors have declared that no competing interests exist.

**Abbreviations:** Leu, Leucine; Phe, Phenylalanine; Ile, Isoleucine; Asn, Asparagine; Lys, Lysine; Tyr, Tyrosine; Val, Valine; Ala, Alanine; Pro, Proline; Arg, Arginine; Gln, Glutamine; Arg, Arginine; Gly, Glycine; Glu, Glutamic acid; Asp, Aspartic acid; Cys, Cysteine; His, Histidine; Met, Methionine; Ser, Serine; Trp, Tryptophan; PCG, Protein-coding genes; RSCU: Relative synonymous codon usage; tRNAs: transfer RNAs; rRNAs: ribosomal RNA; BP,Bootstrap proportion; PP, Posterior probability

the Macroheterocera [3,4]. The clade Macroheterocera consist of six superfamilies (Bombycoidea, Lasiocampoidea, Geometroidea, Noctuoidea, Drepanoidea and Mimallonoidea) [3]. The phylogenetic position of the superfamilies Mimallonoidea and Drepanoidea is highly contentious [5]. Most analyses placed the superfamily Mimallonoidea as the basal to other macroheteroceran taxa followed by Drepanoidea [4,6–8]. Consequently, the superfamily Drepanoidea as sister to other Macroheteroceran superfamilies is confirmed [9–11]. Additionally, the phylogenetic positions of superfamilies such as Drepanoidea and Lasiocampoidea have also been provided [8,12]. The majority of studies established the relationships between the other macroheteroceran superfamilies Noctuoidea + (Geometroidea + (Bombycoidea + Lasiocampoidea)) based on numerous data, for instance mitogenome sequences [4,13,14], 19 protein-coding nuclear genes [7], 741 genes from transcriptome data [6]. However, based on transcriptome sequences from 2696 genes, [15] proposed an alternative phylogenetic relationship, viz. ((Noctuoidea + Geometroidea) + (Bombycoidea + Lasiocampoidea)). Inferences of the phylogenetic relationships within the Macroheterocera (Big Moths) have been intensively discussed based on the numbers of molecular markers. Nevertheless, the relationships between these superfamilies have not yet been satisfactorily resolved. It seems to be that the use of different molecular markers and inadequate data led to these phylogenetic variations in Macroheterocera.

The insect mitochondrial genome (mitogenome) is a circular and double-stranded molecule that usually contains 13 protein-coding genes (PCGs), two ribosomal RNA genes, 22 transfer RNA genes (tRNAs) [16–18], and additionally a large non-coding control region known as the A+T-rich region that gets involved in transcription and replication initiation [19–21]. The A+T-rich region comprises numerous polyadenine (polyA) and polythymine (polyT) motifs, repetitive elements, and low complexity sequences found in insect mitogenomes. This composition contributes to the understanding of mtDNA transcription and its regulatory mechanisms in invertebrates [22]. Mitochondrial genome has been steadily used to address the complex phylogenetic questions, as it is characterized by smaller size, cellular richness, non-appearance of introns, rapid evolution, lack of extensive recombination [23].

Mitogenome sequences have gained extensive use in the evolution and phylogeny of insect genomes because they include strong phylogenetic information [4]. Several studies have used mitogenomes to analyse the phylogenetic relationships of moths, such as Geometroidea, Noctuoidea and Bombycoidea. Even though mitogenomes of about 230 macroheteroceran species have been sequenced, comparative investigation of macroheteroceran mitogenomes is rarely conducted [4]. In addition, most lepidopteran phylogenetic analyses used only 13 mitochondrial PCGs [18,24,25], or used a small number of taxa from the Macroheterocera clade [13,14,26]. An innovative study was conducted by Chen *et al.,* [27] who made a thorough phylogenetic analysis within Lepidoptera using concatenated datasets, comprising 13 PCGs.

The Noctuoidea, also named "Owlet moths", is the largest macroheteroceran superfamily and has 42,407 described species [1]. The Noctuoid caterpillars are phytophagous and many species, including armyworms (*Spodoptera* spp.), bollworms (*Helicoverpa* spp.), and cutworms (*Agrotis* spp.), are pests of major importance

in agricultural crops and forestry plants [14,28]. The typical apomorphic character with a metathoracic tympanal organ provided morphological evidence for Noctuoidea's monophyly [29], which was also strongly confirmed by a number of molecular datasets [7,9,30]. However, the phylogeny inconsistencies within Noctuoidea between morphological and molecular results were common [31]. Oenosandridae, Doidae, Notodontidae, Lymantriidae, Arctiidae, Aganaidae and Noctuidae were the seven families recognized by Miller [29]. Later, the families Arctiidae, Lymantriidae were downgraded to subfamily status within the huge family Erebidae [30]. Despite the complexity of Noctuoidea's taxonomic history, six families are currently recognized: Oenosandridae, Notodontidae, Erebidae, Euteliidae, Nolidae and Noctuidae [1,14,30]. The superfamily Noctuoidea has presented challenging phylogenetic problems for a long time [32]. Past three decades have seen significant advancements in resolving issues, in part due to the development of molecular phylogenetics studies [29,30,32–43]. Despite numerous investigations into the intricate genetic relationships among Noctuoidea, no clear conclusion has been reached.

This study is to reconstruct the phylogenetic tree of clade Macroheterocera at the superfamily, family, subfamily, and tribal level relationships. Observing some structural similarities of the superfamily from various elements of the mitogenome, PCGs, tRNAs, rRNAs, and A+T-rich region is the purpose of this analysis. This inference would offer basic evolutionary information on macroheteroceran mitogenomes. Additionally, it is essential to incorporate more data from each taxon to develop a comprehensive phylogenetic tree and resolve the uncertainty of the macroheteroceran clade.

Currently, the mitogenomes of Noctuoidea have been reported in GenBank, totalling nearly 220 (https://www.ncbi.nlm. gov/). This represents less than one percent of the described species within the Noctuoid superfamily. Increasing number of complete mitogenomes for the phylogenetic analysis of Noctuoidea will provide additional evidence, thereby enhancing our comprehension of the phylogenetic relationships within the Noctuoidea superfamily in relation to other macroheteroceran superfamilies. In this analysis, we present a comprehensive molecular phylogenetic analysis of macroheterocera, combining five newly sequenced and annotated the mitogenomes of *Episparis tortuosalis*, *Pandesma quenavadi*, *Erebus macrops, Polydesma boarmoides* and *Xanthodes albago* along with previously available public data to establish a robust evolutionary framework.

## Materials and methods

### Sample collection and genomic DNA extraction

The specimens of five Noctuoidea moths such as *E. tortuosalis* (10º 27′ 04″N 77º 53′ 36″E), *P. quenavadi* (10º 23′ 915″N 77º 49′ 7716″E), *E. macrops* (10º 23′ 5367″N 77º 49′ 2933″E), *P. boarmoides* (10º 23′ 5367″ N 77º 49′ 2933″E) and *X. albago* (10º 27′ 04″N 77º 53′ 36″E) were collected using light trap from the Tamil Nadu part of the Western Ghats. The species are not collected from reserve forest areas and not endangered, so specific permission is not required for this study.

The species were identified, immediately preserved in 100% ethanol, and stored at −80°C in the laboratory for DNA extraction. The genomic DNA was isolated from the thorax tissue of adult moths using Quick-DNA Tissue/Insect Microprep Kit (Cat No-D6016-HSN CODE-38220090, Zymo Research, USA), according to the manufacturer's procedures. A NanoDrop1000 spectrophotometer was used to measure the purity of the DNA samples, and a 1% agarose gel was used to confirm the results.

### Mitogenome sequencing

The samples were undergone quality-check for library preparation. Truseq Nano library preparation kit (Illumina #20015964) was used to make an indexed library from 100 ng of DNA. Subsequent libraries were quantified using a DNA HS assay kit (Thermofisher #Q32851) in accordance with the manufacturer's protocol using a Qubit 4.0 fluorometer (Thermofisher #Q33238). We queried the library using highly sensitive D100 screen tapes (Agilent # 5067–5581) in accordance

with manufacturer's instructions to determine the library size. Molsys Scientific Pvt. Ltd. (Bangalore, India) carried out the next-generation sequencing. Finally, 150 bp paired-end reads were sequenced using the NOVASEQ 6000 platform (Illumina, San Diego, California, USA) at a depth of around 4 GB per sample.

## Sequence assembly and annotation

The NOVOPLASTY Ver 4.2 (https://github.com/ndierckx/NOVOPlasty) was used for the mitochondrial genome assembly [44]. The *de novo* assembler NOVOPLASTY is the only one that provides a quick and simple way to extract extranuclear genomes from whole genome sequencing (WGS) data, producing a single circular contig of excellent quality. Sequence annotations were performed using the MITOS and MITOS2 (http://mitos2.bioinf.unileipzig.de/index.py) webservers [45]. The MITOchondrial Genome Annotation Server (MITOS) is an entirely automated system designed for the *de novo* annotation of metazoan mitochondrial genomes. Moreover, the annotated sequences were once again validated for the precise spans of the 13 protein-coding genes and certain tRNA genes using CHLOROBOX-GeSeq-Annotation of Organellar Genomes (https://chlorobox.mpimp-golm.mpg.de/geseq.html) [46].

## Comparative analysis

NCBI BLAST was used to confirm the overall lengths of the translated amino acid sequences of the PCGs genes. AT skew=[AT]/[A+T] and GC skew=[GC]/[G+C] formulae were used to calculate the skewness compositions (https://en.vectorbuilder.Com/tool/gc-content-calculator). The tRNA genes and their putative secondary structures were predicted using MITOS2 software and analyzed by comparison with the nucleotide sequence of other lepidopteran tRNA sequences. MITOS2 software was used to predict the tRNA genes and their putative secondary structures. The analysis involved comparing the anticipated structures to the nucleotide sequences of other lepidopteran tRNA sequences. Tandem repeats were identified in the A+T-rich using the Tandem Repeats Finder tool (http://tandem.bu.edu/trf/trf.html). Manual calculations were done to calculate the overlapping regions and intergenic spacers between genes. MEGA X was used to calculate the Relative Synonymous Codon Usage (RSCU) of PCGs [47]. The OGDRAW-Draw Organelle Genome Maps (https://chlorobox.mpimp-golm.mpg.de/OGDraw.html) was used to create the circular mitogenome maps [48]. DnaSp6.0 was used to examine both synonymous ($dS$) and nonsynonymous ($dN$) substitutions within each PCGs [49].

## Phylogenetic analyses

The phylogenetic trees for *E. tortuosalis, P. quenavadi, E. macrops, P. boarmoides* and *X. albago* from two different families were reconstructed by aligning sequences of the 13 protein-coding genes along with the 225 species representing five following superfamilies of Macroheterocera: Noctuoidea (191 species), Bombycoidea (15 species), Geometroidea (13 species), Lasiocampoidea (4 species) and Drepanoidea (2 species). For this investigation, two outgroup mitogenomes belonging to the superfamily Papilionoidea namely *Papilio polytes* and *Trogonoptera brookiana* were employed (S1 Table). The superfamily Papilionoidea and macroheteroceran superfamilies are monophyletic groups [50].

AFFT was used to align and concatenate the amino acid sequences of the 13 PCGs genes for phylogenetic analysis [51]. Additionally, the alignment of amino acid sequences was imported into Gblocks in order to find and remove sections that were ambiguously aligned. The concatenated sequences were to create the PHYLIP and NEXUS formats, which were used to reconstruct phylogenetic trees. The model-based Maximum Likelihood [52] was used for the phylogenetic tree reconstruction via IQ-TREE in the PhyloSuite V1.2.2 software package (https://github.com/dongzhang0725/PhyloSuite). Based on 5000 bootstrap replications, the appropriate model General Reversible Mitochondrial (mtREV) Gamma distributed with invariant sites (G+I) was used to infer the evolutionary relationships. The best dataset partitioning models and techniques as determined by PartitionFinder2 were used. Using the best model (GTR+I+G) with Invgamma rate variation across variable sites, MrBayes 3.2.6 was used to run a Bayesian inference (BI) analysis on the dataset. The analysis exhibited adequate convergence to ensure the 0.05 average standard deviation for the split frequencies. The BI analysis

was run using one million generations of MCMC, three hot chains, one cold chain, sampling every 1000 generations, and a burn-in of 25% of sampled values. The reconstructed phylogenetic trees were visualized and edited using the FigTree v.1.4.4 [53] (http://tree.bio.ed.ac.uk/software/figtree/) software.

## Results and discussion

### Mitogenome structure, organization and base composition

The newly sequenced five mitogenomes were involved in two subfamilies. All of them were assembled with an entirely complete mitogenome. The complete mitogenome sequences of the five Noctuoidea species have lengths of 15,419 bp for *E. tortuosalis*, 15,514 bp for *P. quenavadi,* 15,869 bp for *E. macrops,* 15,904 bp for *P. boarmoides* and 15,361 bp for *X. albago* and their corresponding GenBank accession numbers are MW879209, MW899034, MW924115, MW969649 and MW813975 respectively (S1 Table). The typical circular structure mitochondrial genome arrangements of the five Noctuoid species are comparable to those of other lepidopterans and consist of 13 protein-coding genes (PCGs), 22 tRNAs, and two rRNA genes, along with a A+T-rich region (Fig 1). Among them, a total of 23 genes encoded on the J- strand, including 9 PCGs (*cox1–3, atp8, atp6, nad2, nad3 nad6*, and *cob*), and 14 tRNA genes (*trnM, trnI, trnW, trnL2, trnK, trnD, trnG, trnA, trnR, trnN, trnS1, trnE, trnT,* and *trnS2*). The other 14 genes (4 PCGs, 8 tRNA and 2 rRNAs) encoded on the N-strand (Tables 1 and 2). The 37 gene lengths, however, did not significantly vary among the five Noctuoid species and other lepidopteran species. The organization of the newly sequenced mitogenomes from five different species is arranged in circular patterns (Fig 1).

All new sequences displayed a high AT nucleotide bias, with A+T% of the whole mitogenome of 81.85% in *E. tortuosalis*, 81.23% in *P. quenavadi,* 80.94% in *E. macrops,* 80.88% in *P. boarmoides*, and 81.73% in *X. albago*. The AT-skews of mitogenomes showed negative values in *E. tortuosalis* (−0.012), *P. quenavadi* (−0.030)*, E. macrops* (−0.034)*, P. boarmoides* (−0.001), except for positive skews in *X. albago* (0.012). The GC-skews showed negative values for all the sequenced mitogenomes (Table 3; S2 Table). We also examined the Noctuoid mitogenome sizes and nucleotide compositions of the earlier sequenced mitogenomes. The size of the A+T-rich region had a major role in determining the changed size of noctuoid's mitochondrial sequences. Amongst all sequenced Noctuoid mitogenome (S1 Table) [4,14,54–69], the size of the mitogenomes of *Oraesia emarginata* (16,668 bp) was the lengthiest due to the longest A+T-rich region (887 bp), whereas that of *Heliothis assulta* (15,183 bp) was the shortest A+T-rich region (173 bp).

### Structural characteristics of the protein-coding genes and codon usage

The 13 protein-coding genes are organized in a stable pattern, comparable to other insects [70,71]. The protein-coding genes in the five mitogenomes are of similar size, and the longest gene *nad5* had an average length of 1737 base pairs. The shortest gene was *atp8*, which had an average length of 164 bp. The average lengths of other protein-coding genes were 1014 bp (*nad2*), 1532 bp (*cox1*), 682 bp (*cox2*), 678 bp (*atp6*), 789 bp (*cox3*), 354 bp (*nad3*), 1338 bp (*nad4*), 291 bp (*nad4L*). 531 bp (*nad6*),1153 bp (*cob*) and 939 bp (*nad1*) (Tables 2 and 3).

In total, thirteen PCGs were noticed, of which nine PCGs (*cox1, cox2, cox3, atp8, atp6, nad2, nad3, nad6* and *cob*) were encoded by the J-strand; however, four (*nad4, nad5, nad4L* and *nad1*) were encoded by N-strand. The PCGs locations and orientations of the five newly sequenced mitogenomes were consistent with the majority of lepidopteran moths (Table 1). Total lengths of PCGs in five Noctuoid species such as *E. tortuosalis* (11,198 bp), *P. quenavadi* (11,197 bp)*, E. macrops* (11,212 bp)*, P. boarmoides* (11,206 bp), and *X. albago* (11,210 bp) (S2 Table), exhibited conservation in size and structures. The A+T contents were 81.85% in *E. tortuosalis,* 80% in *E. macrops,* 81.23% in *P. quenavadi,* 80.88% in *P. boarmoides* and 81.73% in *X. albago* (Table 3; S2 Table). All AT skew values were negative, whereas the value of *X. albago* was positive. Negative GC skews were found in all the PCGs; this was the same as in the other Noctuoid mitogenomes that had been sequenced [54,59,72–78].

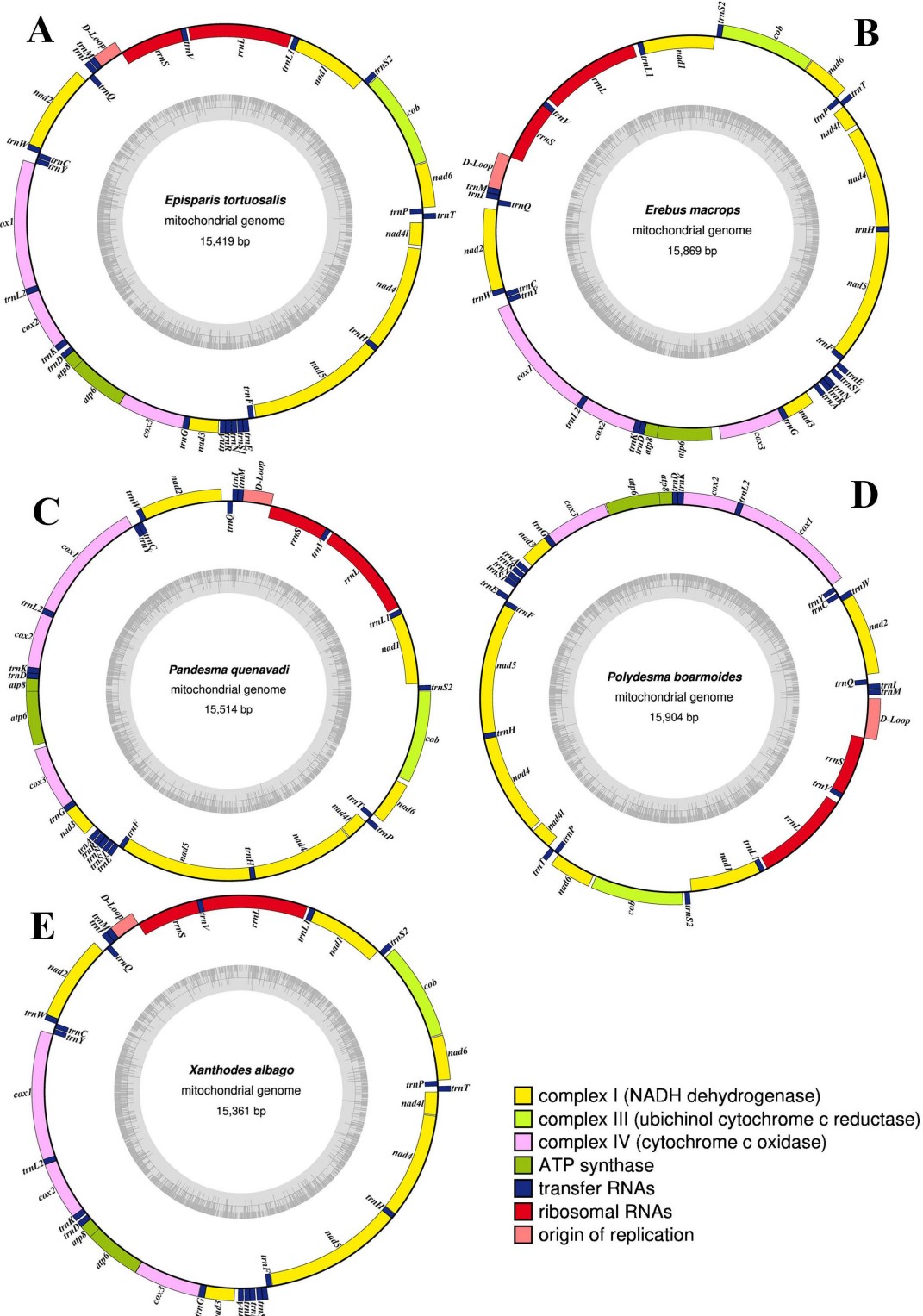

**Fig 1. Circular maps of the newly sequenced complete mitochondrial genomes of A) Episparis tortuosalis, B) Erebus macrops, C) Pandesma quenavadi, D) Polydesma boarmoides, E) Xanthodes albago.**

**Table 1. Details on gene organization of two newly sequenced *Episparis tortuosalis* and *Xanthodes albago* mitogenomes. J and N refer to the majority and minority strands respectively.**

| Gene/strand | Position from | To | Length | Anticodon | Start codon | Stop codon | Intergenic nucleotides |
|---|---|---|---|---|---|---|---|
| *Episparis tortuosalis* and *Xanthodes albago* | | | | | | | |
| trnT/J | 40/5 | 104/70 | 65/66 | TGT | | | 0/0 |
| trnP/N | 105/71 | 169/136 | 65/66 | TGG | | | 7/7 |
| nad6/J | 177/144 | 707/674 | 531/531 | | ATT/ATT | TAA/TAA | 11/11 |
| cob/J | 719/686 | 1870/1837 | 1152/1152 | | ATG/ATG | TAA/TAA | 12/31 |
| trnS2/J | 1883/1869 | 1950/1936 | 68/68 | TGA | | | 18/19 |
| nad1/N | 1969/1956 | 2907/2894 | 939/939 | | ATG/ATG | TAA/TAA | 1/1 |
| trnL1/N | 2909/2896 | 2979/2966 | 71/71 | TAG | | | 32/32 |
| rrnL/N | 3012/2999 | 4312/4338 | 1301/1340 | | | | 28/10 |
| trnV/N | 4341/4329 | 4409/4395 | 69/67 | TAC | | | 0/0 |
| rrnS/N | 4410/4396 | 5197/5181 | 788/786 | | | | 0/0 |
| A+T-rich Region | 5198/5182 | 5498/5483 | 301/302 | | | | 0/0 |
| trnM/J | 5499/5484 | 5566/5551 | 68/68 | CAT | | | 6/0 |
| trnI/J | 5573/5552 | 5640/5619 | 68/68 | GAT | | | −3/-3 |
| trnQ/N | 5638/5617 | 5706/5685 | 69/69 | TTG | | | 57/54 |
| nad2/J | 5764/5740 | 6777/6753 | 1014/1014 | | ATT/ATT | TAA/TAA | −2/-2 |
| trnW/J | 6776/6752 | 6843/6819 | 68/68 | TCA | | | −8/-8 |
| trnC/N | 6836/6812 | 6899/6878 | 64/67 | GCA | | | 7/8 |
| trnY/N | 6907/6887 | 6971/6951 | 65/65 | GTA | | | 6/2 |
| cox1/J | 6978/6954 | 8509/8489 | 1532/1536 | | CGA/CGA | TT/TAA | −1/-5 |
| trnL2/J | 8509/8485 | 8575/8551 | 67/67 | TAA | | | 0/0 |
| cox2/J | 8576/8552 | 9257/9236 | 682/685 | | ATG/ATG | T/T | 0/-3 |
| trnK/J | 9258/9234 | 9328/9304 | 71/71 | CTT | | | 46/21 |
| trnD/J | 9375/9326 | 9440/9394 | 66/69 | GTC | | | 0/0 |
| atp8/J | 9441/9395 | 9605/9556 | 165/162 | | ATT/ATT | TAA/TAA | −7/-7 |
| atp6/J | 9599/9550 | 10276/10227 | 678/678 | | ATG/ATG | TAA/TAA | −1/-1 |
| cox3/J | 10276/10227 | 11064/11015 | 789/789 | | ATG/ATG | TAA/TAA | 3/3 |
| trnG/J | 11068/11019 | 11134/11087 | 67/69 | TCC | | | 1/0 |
| nad3/J | 11135/11088 | 11488/11441 | 354/354 | | ATT/ATT | TAA/TAA | 18/43 |
| trnA/J | 11507/11485 | 11573/11550 | 67/66 | TGC | | | −1/10 |
| trnR/J | 11573/11561 | 11640/11625 | 68/65 | TCG | | | 1/-1 |
| trnN/J | 11642/11625 | 11707/11691 | 66/67 | GTT | | | 8/16 |
| trnS1/J | 11716/11708 | 11781/11773 | 66/66 | GCT | | | 0/0 |
| trnE/J | 11782/11774 | 11850/11839 | 69/66 | TTC | | | 10/1 |
| trnF/N | 11861/11841 | 11926/11907 | 66/67 | GAA | | | 25/5 |
| nad5/N | 11952/11913 | 13676/13652 | 1725/1740 | | ATA/ATT | TTT/TAA | 0/0 |
| trnH/N | 13677/13653 | 13743/13719 | 67/67 | GTG | | | 0/3 |
| nad4/N | 13744/13723- | 15082/15058 | 1340/1336 | | AAT/ATG | TT/T(AA) | 39/9 |
| nad4l/N | 15123/15068 | 15410/15361 | 297/294 | | TAA/ATG | TAA/TAA | 0/0 |

All PCGs in the mitogenomes of these five species began with conventional initiation codons ATN (ATT, ATA, ATG and ATC) except for *cox1* which used uncommon initiation codon CGA in *E. tortuosalis*, *X. albago*, *P. quenavadi*, *P. boarmoides*. AAT and TAA were also the initiation codons of *nad4* and *nad4l* in *E. tortuosalis* (as annotated by CHLOROBOX;

**Table 2. Details on gene organization of three newly sequenced *Pandesma quenavadi*, *Erebus macrops* and *Polydesma boarmoides* mitogenomes. J and N refer to the majority and minority strands respectively.**

| Gene/strand | Position from | To | Length | Anticodon | Start codon | Stop codon | Intergenic nucleotides |
|---|---|---|---|---|---|---|---|
| *Pandesma quenavadi*, *Erebus macrops* and *Polydesma boarmoides* | | | | | | | |
| trnS2/J | 6/3590/12065 | 71/3655/12130 | 66/66/66 | TGA | | | 21/21/21 |
| nad1/N | 93/3677/12152 | 1031/4615/13090 | 939/939/939 | | ATG/ATG/ATG | TAA/TAA/TAA | 1/17/1 |
| trnL1/N | 1033/4618/13092 | 1100/4686/13159 | 68/69/68 | TAG | | | 33/57/34 |
| rrnL/N | 1134/4744/13194 | 2438/6050/14458 | 1305/1307/1265 | | | | 35/43/78 |
| trnV/N | 2474/6094/14537 | 2539/6161/14602 | 66/68/66 | TAC | | | −1/0/0 |
| rrnS/N | 2539/6162/14603 | 3319/6948/15387 | 781/787/785 | | | | 0/0/0 |
| A+T-rich Region | 3320/6949/15388 | 3692/7390/15904 | 373/442/517 | | | | 0/0/0 |
| trnM/J | 3693/7391/46 | 3760/7457/112 | 68/67/67 | CAT | | | 0/1/5 |
| trnI/J | 3761/7459/118 | 3827/7525/184 | 67/67/67 | GAT | | | −3/-3/-3 |
| trnQ/N | 3825/7523/182 | 3893/7591/250 | 69/69/69 | TTG | | | 73/59/62 |
| nad2/J | 3967/7650/313 | 4980/8663/1326 | 1014/1014/1014 | | ATT/ATT/ATT | TAA/TAA/TAA | 0/-2/3 |
| trnW/J | 4981/8662/1330 | 5049/8728/1398 | 69/67/69 | TCA | | | −8/-7/-8 |
| trnC/N | 5042/8721/1391 | 5107/8788/1454 | 66/68/64 | GCA | | | −1/10/26 |
| trnY/N | 5107/8799/1481 | 5172/8866/1546 | 66/68/66 | GTA | | | 12/20/32 |
| cox1/J | 5185/8887/1579 | 6714/10420/3109 | 1530/1534/1531 | | CGA/ATG/CGA | AAC/T/T | 1/0/0 |
| trnL2/J | 6716/10421/3110 | 6782/10487/3176 | 67/67/67 | TAA | | | 0/0/0 |
| cox2/J | 6783/10488/3177 | 7464/11169/3858 | 682/682/682 | | ATA/ATA/ATA | T/T/T | 0/0/0 |
| trnK/J | 7465/11170/3859 | 7535/11240/3929 | 71/71/71 | CTT | | | −1/1/-1 |
| trnD/J | 7535/11242/3929 | 7602/11307/3996 | 68/66/68 | GTC | | | 0/0/0 |
| atp8/J | 7603/11308/3997 | 7764/11472/4161 | 162/165/165 | | ATT/ATC/ATT | TAA/TAA/TAA | 7/-7/-7 |
| atp6/J | 7758/11466/4155 | 8435/12143/4832 | 678/678/678 | | ATG/ATG/ATG | TAA/TAA/TAA | 49/104/10 |
| cox3/J | 8485/12248/4843 | 9273/13036/5631 | 789/789/789 | | ATG/ATG/ATG | TAA/TAA/TAA | 2/2/2 |
| trnG/J | 9276/13039/5634 | 9341/13104/5699 | 66/66/66 | TCC | | | 0/0/0 |
| nad3/J | 9342/13105/5700 | 9695/13458/6053 | 354/354/354 | | ATT/ATT/ATT | TAA/TAA/TAA | 62/120/58 |
| trnA/J | 9758/13579/6112 | 9823/13642/6180 | 66/64/69 | TGC | | | 8/33/16 |
| trnR/J | 9832/13676/6197 | 9899/13739/6261 | 68/64/65 | TCG | | | 0/1/19 |
| trnN/J | 9900/13741/6281 | 9966/13806/6346 | 67/66/66 | GTT | | | 8/63/4 |
| trnS1/J | 9975/13870/6351 | 10040/13935/6416 | 66/66/66 | GCT | | | 8/33/124 |
| trnE/J | 10049/13969/6541 | 10115/14035/6608 | 67/67/68 | TTC | | | 1/25/2 |
| trnF/N | 10117/14061/6611 | 10184/14128/6677 | 68/68/67 | GAA | | | −1/0/0 |
| nad5/N | 10184/14129/6678 | 11922/15869/8418 | 1739/1741/1741 | | ATA/ATA/ATA | TA/T/T | 0/0/0 |
| trnH/N | 11923/1/8419 | 11990/68/8485 | 68/68/67 | GTG | | | 0/0/2 |
| nad4/N | 11991/69/8488 | 13329/1407/9826 | 1339/1339/1339 | | ATG/ATG/ATG | T/T/T | 14/55/36 |
| nad4l/N | 13344/1463/9863 | 13631/1750/10150 | 288/288/288 | | ATG/ATG/ATG | TAA/TAA/TAA | 8/5/27 |
| trnT/J | 13640/1756/10178 | 13704/1820/10242 | 65/65/65 | TGT | | | 0/0/0 |
| trnP/N | 13705/1821/10243 | 13769/1885/10308 | 65/65/66 | TGG | | | 7/7/7 |
| nad6/J | 13777/1893/10316 | 14307/2423/10846 | 531/531/531 | | ATT/ATT/ATC | TAA/TAA/TAA | 55/10/36 |
| cob/J | 14363/2434/10883 | 15514/3591/12037 | 1152/1158/1155 | | ATG/ATG/ATG | TAA/TAA/TAA | 0/-2/27 |

[46]). The initiation codons AAT and TAA of *nad4* and *nad4l* in *E. tortuosalis* are not standard mitochondrial initiation codons in invertebrates and echinoderms, although it is a possible start codon in *Escherichia coli* [79,80]. Most PCGs (*cox3*, *atp6*, *atp8*, *nad1*, *nad2*, *nad3*, *nad6*, *nad4l* and *cob*) have typical stop codon (TAA), with the exception of

**Table 3. Composition and Skewness of _Episparis tortuosalis, Pandesma quenavadi, Erebus macrops, Polydesma boarmoides_ and _Xanthodes albago_.**

| Species | Size (bp) | A % | G% | T% | C% | A+T% | AT skew | GC skew |
|---|---|---|---|---|---|---|---|---|
| **_Episparis tortuosalis_** | | | | | | | | |
| Whole Genome | 15,419 | 40.42 | 7.56 | 41.43 | 10.58 | 81.85 | −0.012 | −0.178 |
| PCG | 11,198 | 39.86 | 8.32 | 40.69 | 11.13 | 80.55 | −0.010 | −0.144 |
| tRNA | 1480 | 41.42 | 8.18 | 40.27 | 10.14 | 81.69 | 0.014 | −0.107 |
| rrna | 2089 | 41.31 | 4.98 | 43.56 | 10.15 | 84.87 | −0.026 | −0.341 |
| A+T-rich region | 301 | 43.85 | 2.33 | 50.17 | 3.65 | 94.02 | −0.067 | −0.222 |
| **_Pandesma quenavadi_** | | | | | | | | |
| Whole Genome | 15,514 | 39.39 | 7.54 | 41.84 | 11.23 | 81.23 | −0.030 | −0.196 |
| PCG | 11,197 | 38.95 | 8.27 | 40.88 | 11.9 | 79.83 | −0.024 | −0.180 |
| tRNA | 1477 | 40.42 | 8.33 | 40.83 | 10.43 | 81.25 | −0.005 | −0.111 |
| rrna | 2086 | 40.75 | 5.03 | 43.53 | 10.69 | 84.28 | −0.032 | −0.359 |
| A+T-rich region | 373 | 40.21 | 3.75 | 50.67 | 5.36 | 90.88 | −0.115 | −0.176 |
| **_Erebus macrops_** | | | | | | | | |
| Whole Genome | 15,869 | 39.08 | 7.44 | 41.86 | 11.63 | 80.94 | −0.034 | −0.220 |
| PCG | 11,212 | 38.34 | 8.73 | 40.58 | 12.7 | 78.92 | −0.028 | −0.205 |
| tRNA | 1471 | 40.13 | 8.16 | 40.38 | 10.33 | 81.51 | 0.009 | −0.117 |
| rrna | 2094 | 40.88 | 4.97 | 43.74 | 10.41 | 84.62 | −0.033 | −0.354 |
| A+T-rich region | 442 | 42.53 | 2.49 | 50.0 | 4.98 | 92.53 | −0.080 | −0.333 |
| **_Polydesma boarmoides_** | | | | | | | | |
| Whole Genome | 15,902 | 39.42 | 7.35 | 41.46 | 11.77 | 80.88 | −0.001 | −0.230 |
| PCG | 11,206 | 39.2 | 8.26 | 39.83 | 12.71 | 79.03 | −0.007 | −0.211 |
| tRNA | 1473 | 40.6 | 8.15 | 40.8 | 10.45 | 81.4 | −0.002 | −0.124 |
| rrna | 2050 | 40.59 | 5.07 | 42.93 | 11.41 | 83.52 | −0.028 | −0.384 |
| A+T-rich region | 517 | 40.97 | 2.51 | 52.61 | 2.9 | 94.58 | −0.112 | −0.071 |
| **_Xanthodes albago_** | | | | | | | | |
| Whole Genome | 15,361 | 40.37 | 7.64 | 41.36 | 10.62 | 81.73 | 0.012 | −0.162 |
| PCG | 11,210 | 39.9 | 8.42 | 40.54 | 11.14 | 80.44 | −0.007 | −0.139 |
| tRNA | 1483 | 41.07 | 8.02 | 40.73 | 10.18 | 81.8 | 0.002 | −0.118 |
| rrna | 2126 | 40.97 | 4.99 | 43.74 | 10.3 | 84.71 | −0.032 | −0.347 |
| A+T-rich region | 302 | 45.36 | 1.32 | 49.34 | 3.97 | 94.7 | −0.064 | −0.5 |

incomplete terminal codons like T used for _cox2_ (_E. tortuosalis, X. albago, P. quenavadi, P. boarmoides_ and _E. macrops_) _nad4_ (_P. quenavadi, E. macrops_ and _P. boarmoides_), _cox1_ (_P. boarmoides_ and _E. macrops_), _nad5_ (_P. boarmoides_ and _E. macrops_) as well as TT for _cox1_ and _nad4_ (_E. tortuosalis_), TTT used for _nad5_ (_E. tortuosalis_), AAC for _cox1_ (_P. quenavadi_), and TA for _nad5_ (_P. quenavadi_). The processes of polycistronic transcription cleavage and polyadenylation depend heavily on incomplete terminal codons [81,82] which is not frequently seen in the mitogenomes of Noctuoidea.

The relative synonymous codon usage (RSCU) and the amino acid composition of the five Noctuoid mitogenomes were analyzed and summarized. The comparative analysis of the relative synonymous codon usage reveals obvious bias (Fig 2; S3 Table). The codons that were most commonly utilized UUU (Phe), UUA (Leu), AUU (Ile), AAU (Asn) AAA (Lys) and UAU (Tyr) were the most consistently used codons (>179) in the PCGs of the five mitogenomes whereas CGC (Arg) and GCC (Ala) were the least used codons (<20). The pattern of codon usage in newly sequenced mitogenomes is very similar to that of previously sequenced lepidopteran mitogenomes [4,54,56,60,63–65,83–90].

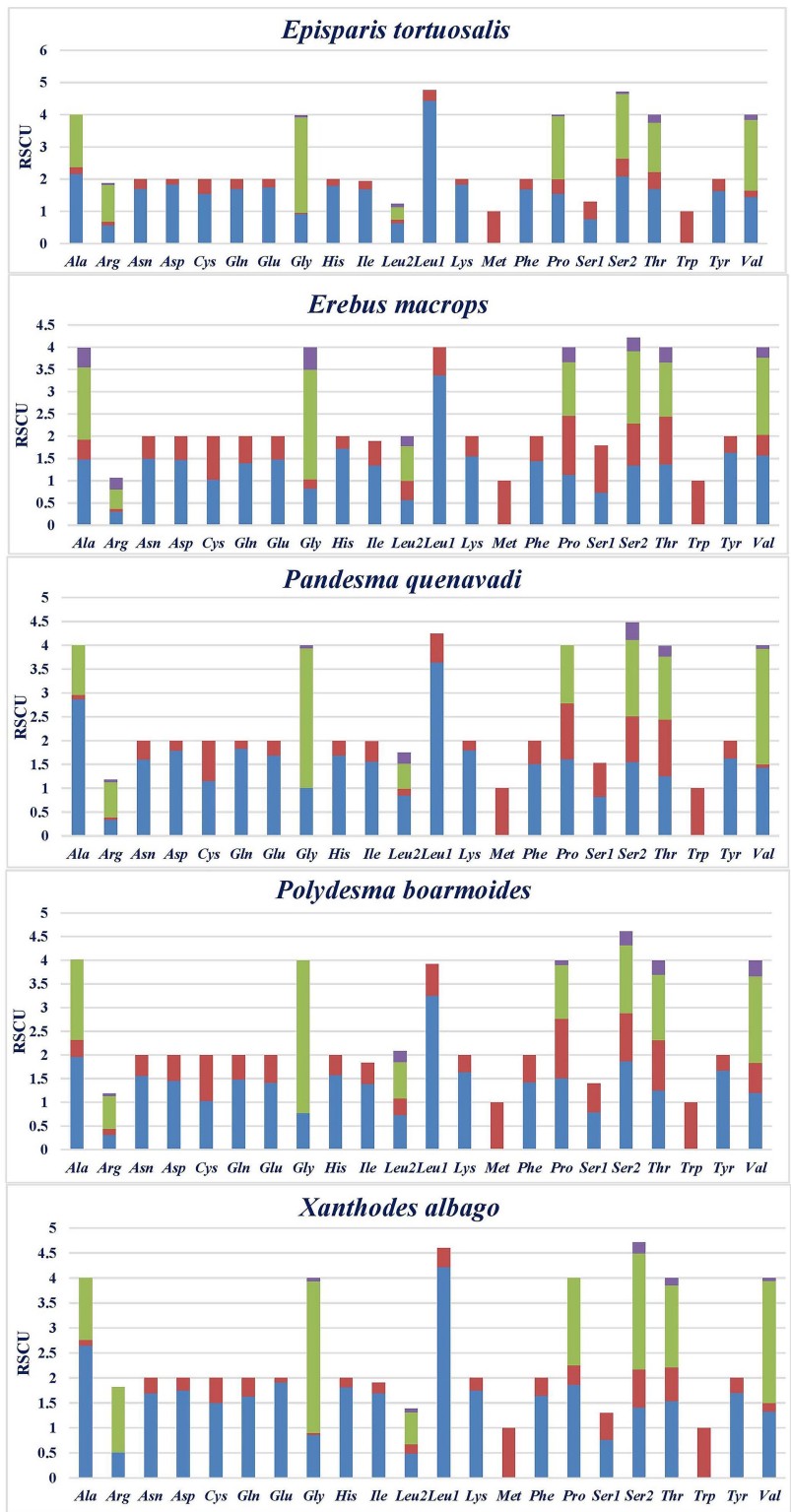

**Fig 2. Relative Synonymous Codon Usage (RSCU) in Noctuoidea PCGs.**

### Synonymous and non-synonymous substitution rates.

The widely accepted Ka/Ks ratio defined the molecular evolutionary relationships and selection pressure between homogeneous and heterogeneous species [91]. According to reports, Ka/Ks > 1 indicates positive selection, Ka/Ks = 1 indicates neutrality, and Ka/Ks < 1 indicates negative selection. The benefit of this approach is that it illustrates the impact of natural selection on PCGs. We calculated the Ka/Ks substitutions and compared them to those of 211 other Noctuoid species to examine the rates of evolution of homologous gene pairs. The values of Ka, Ks, and Ka/Ks were analysed in order to investigate the evolutionary patterns of PCGs. 13 PCGs had average Ka/Ks values ranging from 0.321 (*cox2*) to 1.0759 (*cob*) and resulted in the subsequent descending order: *cob > nad4l > nad6 > cox3 > nad5 > nad2 > cox1 > nad1 > nad 3 > nad4 > atp8 > atp6 > cox2.* The majority of PCGs in these superfamilies have Ka/Ks values less than 1, which suggests a significant negative selection. The average Ka/Ks variation was > 1 in *cob*. The *nad4* and *atp6* showed the highest and lowest evolutionary rates (Fig 3). Remarkably, the Ka/Ks ratios of all of the PCGs are lower than one, indicating that they are evolving under purifying selection and making them suitable for assessing evolutionary relationships within the Macroheterocera.

### Intergenic spacers and overlapping sequences

Insect mitogenomes have several intergenic and gene-overlapping regions. Three conserved gene-overlapping areas were found with nucleotide lengths of 5 bp, 7 bp, and 8 bp in five newly sequenced mitogenomes. The junction of *cox1* and *trnL2* have the "TCTAA" 5 bp conserved gene overlapping region (Fig 4). Between *atp6* and *atp8*, the 7-bp overlap of "ATGATAA", as presented in Fig 5, is commonly found in all lepidopteran mitogenomes [56,92–96]. The 8 bp overlap between *trnC* and *trnW* comprised the motif "AAGCCTTA" (Fig 6). In addition, the intergenic region of "ATACTAA" was detected between *nad1* and *trnS2* among five mitogenomes (Fig 7). The motifs were conventionally extant through-out Macroheterocera mitogenomes [4] and were responsible for mitochondrion transcription [19,97]. In five recently sequenced mitogenomes, the intergenic spacers situated between *nad2* and *trnQ* varied in length from 54 to 73 bp, which is a common feature in all lepidopteran insects.

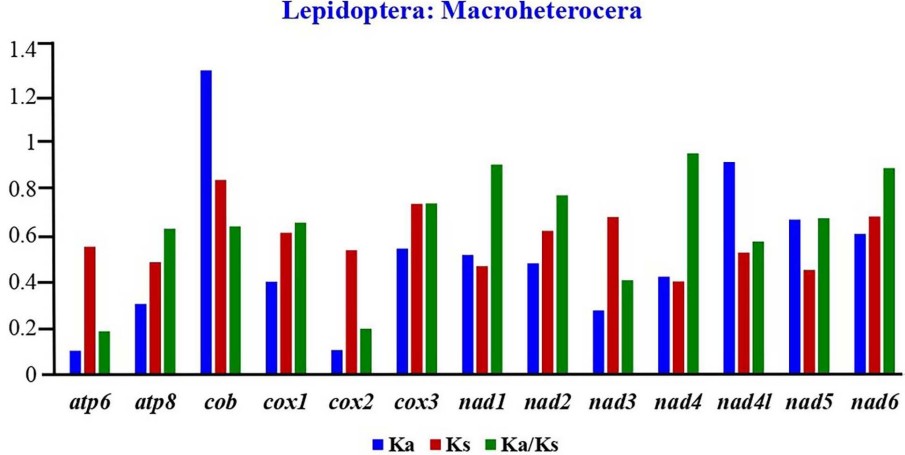

**Fig 3. Rate of non-synonymous substitutions (Ka), rate of synonymous substitutions (Ks), and ratio of rate of non-synonymous substitutions to rate of synonymous substitutions (Ka/Ks).**

|  | *Cox1* |  | *trnL2* |
|---|---|---|---|
| *Episparis tortuosalis* | TTAAGTAACT | **TCTAA** | TATGACAGAT |
| *Erebus macrops* | TTAAGTAATT | **TCTAA** | TATGGCAGAT |
| *Xanthodes albago* | TTAAGTAACT | **TCTAA** | TATGGCAGAT |
| *Pandesma quenavadi* | TTAAGAAACT | **TCTAA** | TATGGCAGAT |
| *Polydesma boarmoides* | TTAAGAAATT | **TCTAA** | TATGGCAGAT |
| *Spodoptera litura* | TTAAGTAACT | **TCTAA** | TATGGCAGAT |
| *Eutelia adulatricoides* | TTAAGTAATT | **TCTAA** | TATGGCAGAT |
| *Ochrogaster lunifer* | AATAATAATT | **TCTAA** | TATGACAGAT |
| *Gabala argentata* | TTAAGTAATT | **TCTAA** | TATGGCAGAT |
| *Doa* sp. MJT-2014 | TTAAATAATT | **TCTAA** | TATGGCAGAA |
| *Kunugia undans* | TTAAGTTATT | **TCTAA** | TATGACAGAC |
| *Bombyx mandarina* | TTAAGAAATT | **TCTAA** | TATGACAGAC |
| *Idaea effusaria* | TTAAGTATAT | **TCTAA** | TATGGCAGAT |

**Fig 4. The intergenic spacer region between *Cox1* and *trnL2*.**

|  | *atp8* |  | *atp6* |
|---|---|---|---|
| *Episparis tortuosalis* | TTTTAATTGAAA | **ATGATAA** | GTAATTTATTTTCAA |
| *Erebus macrops* | TTTAAATTGAAA | **ATGATAA** | GTAATCTATTTTCAA |
| *Xanthodes albago* | TTTATATTGAAA | **ATGATAA** | GTAATTTATTTTCAA |
| *Pandesma quenavadi* | TTTAATTTGAAA | **ATGATAA** | GTAATTTATTTTCAA |
| *Polydesma boarmoides* | TTTATCTTGAAA | **ATGATAA** | GTAATTTATTTTCTA |
| *Eutelia adulatricoides* | TTTTACTTGAAA | **ATGATAA** | GTCAATTTTTGACCC |
| *Spodoptera litura* | TTTTAATTGAAA | **ATGATAA** | GTAATTTATTTTCAA |
| *Ochrogaster lunifer* | ATTTTTTTGAAA | **ATGATAA** | GTAACTTATTTTCAG |
| *Gabala argentata* | TTTTTTTTGAAA | **ATGATAA** | GAAATTTATTTTCTA |
| *Doa* sp. MJT-2014 | AATTTTATGAAA | **ATGATAA** | GTAACTTATTTTCAA |
| *Bombyx mandarina* | TTTTAATTGAAA | **ATGATAA** | CAAACTTATTTTCTA |
| *Idaea effusaria* | TATTAACTGAAA | **ATGATAA** | CTAATTTATTTTCAA |
| *Kunugia undans* | CTTTAATTGAAA | **ATGATAA** | CAAACTTATTTTCTA |

**Fig 5. Alignment of *atp8* and *atp6* overlap among Noctuoid, Bombycoid, Lasiocampoid, Geometroid and Drepanoid species.**

|  | *trnW* |  | *trnC* |
|---|---|---|---|
| *Episparis tortuosalis* | AATTATTCTTT | **AAGCCTTA** | ATAGATTATCTATTCCT |
| *Erebus macrops* | AATTATTCTTT | **AAGCCTTA** | GTATATTATTTACTCCT |
| *Xanthodes albago* | AAATATTCTTT | **AAGCCTTA** | ATAGTAAACTATTCCTT |
| *Pandesma quenavadi* | AATTATTCTTT | **AAGCCTTA** | GTATATTTTATTTACTC |
| *Polydesma boarmoides* | GATTCTTCTTT | **AAGCCTTA** | GTATATTTATTACTCCT |
| *Eutelia adulatricoides* | AATTATTCTTT | **AAGCCTTA** | GTAATATTTATTTACTC |
| *Spodoptera litura* | AAAATTTCTTT | **AAGCCTTA** | ATAGAATTATAACTATT |
| *Gabala argentata* | GAAATTTCTTT | **AAGCCTTA** | GTATATAATGTACTCCT |
| *Doa* sp. MJT-2014 | GAAATTTCTTT | **AAGCCTTA** | GTAAATTTTACTCCTTA |
| *Kunugia undans* | TTCTTTTCTTT | **AAGCCTTA** | AAAATTTTATATTTCCC |
| *Bombyx mandarina* | ATTAAATCTTT | **AAGCCTTA** | ATATTTATTAAAATACT |
| *Idaea effusaria* | AAATATTCTTT | **AAGCCTTA** | AATATTTTTATATTAAC |

**Fig 6. Alignment of *trnW* and *trnC* overlap among Noctuoid, Bombycoid, Lasiocampoid, Geometroid and Drepanoid species..**

|  | trnS2 | intergenic sequence | nad1 |
|---|---|---|---|
| *Episparis tortuosalis* (Noctuoidea, Noctuidae) | TCTATTAATTT | **ATACTAA**AAATAATTAAA | TTATATTAAAAAAATTTTAA |
| *Erebus macrops* (Noctuoidea, Erebidae) | TCTATTAATTT | **ATACTAA**AAATAATCAATAAT | TTATATTAAAAAAATTTTTA |
| *Xanthodes albago* (Noctuoidea, Noctuidae) | TCTATTAATTT | **ATACTAA**AAATAATTTAAA | TTATATTAAAAAAATTTTAA |
| *Pandesma quenavadi* (Noctuoidea, Erebidae) | TCTATTAATTT | **ATACTAA**AATTAATTATTAAA | TTATATTAAAAAAATTTTT |
| *Polydesma boarmoides* (Noctuoidea, Erebidae) | TCTATTAATTT | **ATACTAA**AAATAATCAATAAA | TTACAATAAAAAAATTTTT |
| *Eutelia adulatricoides* (Noctuoidea: Euteliidae) | TCTATTAATTT | **ATACTAA**AAAAAAATCAATTTT | TTAAATTAAAAAAATTTTA |
| *Ochrogaster lunifer* (Noctuoidea, Notodontidae) | TCTATTAATTT | **ATACTAA**AAATAATTAA | TTAAAAAAAAATATAAATTTTAAC |
| *Gabala argentata* (Noctuoidea, Nolidae) | TCTATTAATTT | **TATACTAA**AAATAATTAAT | TTATAATAAAAAAATTTTA |
| *Doa* sp. MJT-2014 (Drepanoidea Drepanidae) | TCTATTAATTT | **ATACTAA**AAATT | TAACATAAAAAAATTTTTA |
| *Bombyx mandarina* (Bombycoidea, Bombycidae) | TCTATTAATTTTTTATTA**ATACTAA**AAATATTACAA | | TTAAAATAAATAAAATTTTAA |
| *Idaea effusaria* (Geometroidea, Geometridae) | TCTATTAATTT | **ATACTAA**AAATAATTTATTATT | TATAAAATAAAAATTTTTA |

**Fig 7. The intergenic spacer region between *trnS2* (UCN) and *nad1* was aligned for Noctuoid, Bombycoid, Lasiocampoid, Geometroid and Drepanoid species.**

## Transfer RNA genes (tRNA)

The traditional 22 tRNAs of the mitogenomes of the noctuoids were found in the five mitogenomes and intermingled throughout the PCGs. Fourteen of these tRNAs were encoded with the majority strand, while the remaining 8 tRNA genes (*trnQ, trnC, trnY, trnF, trnH, trnP, trnL1* (CUN) and *trnV*) were encoded by the minority strand in *E. tortuosalis*, *P. quenavadi, E. macrops, P. boarmoides*, and *X. albago* (Table 1). With the exception of the *trnS1* gene, all tRNA genes could form the conventional cloverleaf secondary structure, which includes the acceptor arm, dihydrouridine (DHU) arm, anticodon arm, and pseudouridine (TΨC) arm, resulting in an incomplete secondary structure. It was considered that the lack of the DHU arm meant that the *trnS1* gene had undergone early Metazoan evolution, which was typical of lepidopteran insects as well as other metazoan mitogenomes [4,98–100]. The tRNA gene lengths of the five Noctuoid species, which ranged from 64 bp (*trnA* of *E. macrops*) to 71 bp, were essentially equivalent (*trnK* of all species). On the tRNAs of the five mitogenomes, there were mismatched base pairs in the DHU arm, Acceptor arm, and anticodon arm, the majority of which were U-G mismatches, followed by G-U mismatches, and infrequently U-U mismatches (S1-S5 Figs) (Table 4).

## Structural characteristics of the Ribosomal RNA genes

Five new mitogenome sequences had two rRNAs; *rrnL* (*16S*rRNA) was positioned between *trnL1* (CUN) and *trn; rrnS* (*12S*rRNA) was positioned between *trnV* and the A+T-rich region. The two rRNA genes (*rrnL* and *rrnS*) were encoded on the N-strand. In the five recently sequenced mitogenomes, the length of the *rrnL* was 1301 bp in *E. tortuosalis,* 1340 bp in *X. albago*, 1305 bp in *P. quenavadi*, 1307 bp in *E. macrops* and 1265 bp in *P. boarmoides*, respectively. The A+T contents of total rRNA genes were 84.87% (*E. tortuosalis*), 84.71% (*X. albago*), 84.28% (*P. quenavadi*), 84.62% (*E. macrops*) and 83.52% (*P. boarmoides*). In these five new sequences, both rRNAs displayed negative values of AT skew and GC skew (Table 3; S2 Table).

**Table 4. Mismatch base pairs among the tRNAs of the five mitogenomes.**

| Species | DHU Arm | | | Acceptor Arm | | TΨC Arm | | Anticodon loop | |
|---|---|---|---|---|---|---|---|---|---|
|  | G-U | U-G | U-U | G-U | U-G | G-U | U-G | G-U | U-G |
| *E. tortuosalis* | 4 | 3 |  | 2 | 4 |  | 2 |  | 2 |
| *X. albago* | 5 | 3 | 1 | 2 | 3 |  | 2 | 1 | 3 |
| *P. quenavadi* | 4 | 4 | 1 | 3 | 3 |  | 4 | 1 | 2 |
| *E. macrops* | 4 | 3 |  | 1 | 3 | 1 | 1 | 2 | 3 |
| *P. boarmoides* | 4 | 3 |  | 2 | 3 |  | 1 | 4 | 2 |

## Structural characteristics of the A+T rich region

The A+T-rich region (control region) is the longest non-coding region of the mitogenomes of these five species and was positioned between *rrnS* and *MIQ* cluster, with full lengths of 301 bp (*E. tortuosalis*), 302 bp (*X. albago*), 373 bp (*P. quenavadi*), 442 bp (*E. macrops*) and 517 bp (*P. boarmoides*); this was considered to be related to the origin of replication and transcription of mitogenomes [17,18,21,82]. The highest proportion of adenine and thymine was found in this region (S2 Table). All the A+T contents showed a clear AT bias, with an AT of 94.02% (*E. tortuosalis*), 92.53% (*E. macrops*), 90.88% (*P. quenavadi*), 94.58% (*P. boarmoides*) and 94.7% (*X. albago*). In the control region of the lepidopteran mitogenomes, three preserved structures were present: the ATAGA motif, the 15–19 bp poly–T stretch, and the poly–A stretch. The motif "ATAGA" followed by poly-T was present at the beginning of the A+T-rich region of newly sequenced mitogenomes (Fig 8). But the motif ATAGA and poly–T disappeared in the mitogenome of *Oraesia emarginata* [95]. In most lepidopteran mitogenomes, the poly-A stretch, the conserved structure can be found near to the 5′ end of *trnM* [27]. Both the AT-skew and GC-skew were negative in the control region of all newly sequenced mitogenomes (Table 3; S2 Table). The longest conserved sequences and tandem repeats (T/A), which were similar to the three structures mentioned above, were also observed in the A+T-rich regions of present analysis. The five newly sequenced mitogenomes' A+T-rich structural information is shown in Fig 9. The A+T-rich area of the majority of insects is generally characterized by the presence of multiple tandem repeat elements [20,101]. The A+T-rich region of *P. quenavadi* had two tandem repeats of 40 bp and 41 bp. Four types of tandem repeats were present in *E. macrops* with small sizes of 48 bp, 55 bp, 63 bp and 70 bp. Moreover, five tandem repeat elements were observed in *P. boarmoides* with sizes of 42 bp, 74 bp, 114 bp and 127 bp. Tandem repeat elements have been recognized in several Noctuoid species [54,63,64]. However, in *E. tortuosalis* and *X. albago* mitogenomes sequenced in this study, no tandem repeat elements were observed. Similarly, the tandem repeat elements were also not present in some Noctuoid mitogenomes like *Hydrillodes repugnalis* [102], *Euproctis similis* [89] and *Dysgonia stuposa* [67]. The A+T-rich region also contains a few short microsatellites, repeat regions for example (TA)n prior to the structural motifs ATTTA and ATATTA that is common to all the lepidopteran mitogenomes. Furthermore, we identified microsatellite-like (TA)n, motifs (ATTTA)n (ATATTA)n in the A+T-rich regions of newly sequenced mitogenomes (Fig 9).

## Phylogenetic analyses of Macroheterocera

The substitution saturation for the combined dataset of the 13 PCG in DAMBE 7 [103] was evaluated, and the *Iss* value (0.0815) was substantially lower than the critical value. Since the combined sequence substitution was unsaturated, the sequences were suitable to perform phylogenetic analysis.

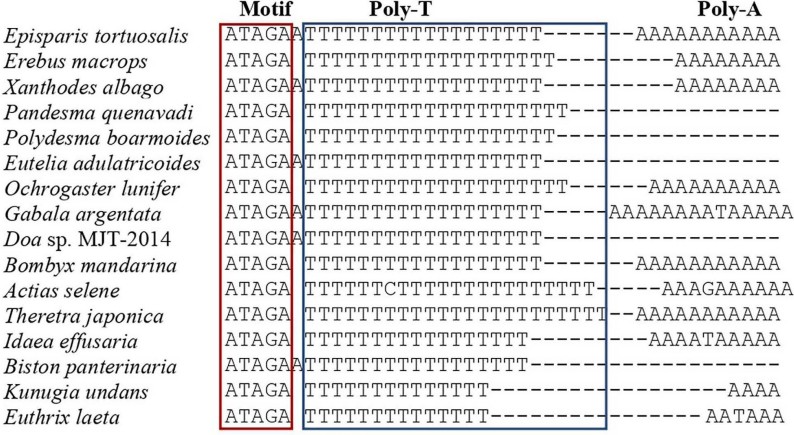

**Fig 8. Alignment of motif ATAGA, poly-T and poly-A in A+T-rich region of Noctuoidea, Bombycoidea, Lasiocampoidea, Geometroidea and Drepanoidea.**

*Episparis tortuosalis*
*rrnS*- **5, 197** TTTAATGTAACATTTTTTTTAA**ATAGA**A**TTTTTTTTTTTTTTTTTTTT**ATATT
AAAAT**ATTTA**A**ATATTA**ATTAATAAAC**ATTTA**ATATTTTATTTTTCTTTCTTTTTATA**AT**
**ATTA**ATATTGAATAC**TATATATATA**TTAAACAGATATAATTCATATAGATTATAATAT**ATA**
**TTA**ATATAATTA**ATATTA**ATTTTTTTAGTT**ATTTA**ATGAATTAATATAA**ATATTA**ATAA
ATTAAAT**ATTTA**A**TATATATATATATATATATA**CAAAACCATTTTTAATAATTTTTC
ATAT**AAAAAAAAAAA**−*trnM*- **5,499**

*Erebus macrops*
*rrnS* -**6, 948** TAAATTTTCTAAACTATAAAAATTTTTATTATTATTATTATATGCATATT
TTTACAC**ATAGA**T**TTTTTTTTTTTTTTTTTTTTT**AT**ATTTA**AAT**ATTTA**ATATA**ATATTA**TA
**TTTT**MATATTAAAAT**ATTTA**ATATAATT**ATTA**AACATTAAATAATCTCTTTTATTTTTTT
TCTCTAATATTC**ATATTA**AATACCAATTTGGAAATTAAACAATGATAATTCTTAAAAAT
TACAAT**ATATTA**AT**ATATT**AAT**ATTA**ATTTTTCTTACTGTTAAGTTATTGAATTTTAAA
T**ATATTA**ATT**ATTTA**AAA**ATTTA**A**TATATATATA**TTAAT**ATTA**AT**TATATA**AAT**ATTTA**
A**ATATATATATATATATA**AAAAATTAAATAATAAA**ATTTA**ATTTTTATGTTGCTAAA
CCATTGTTATTAAATTTTCTTTAAATAT**AAAAAAA**TT−*trnM* -**7, 391**

*Xanthodes albago*
*rrnS* - **5, 181** TTTAAAGATAACTTTTTTTTAA**ATAGA**A**TTTTTTTTTTTTTTTTTTTT**ATATT
AAAAT**ATTTA**ATAATCAATAATAAAT**ATTTA**ATAATTTCTTTTTCTTTCTTTTTATAAT
**ATTA**ATATTA**AATACAAAATAAATAATAAACAATTATAATTCATATAGATAA**ATATTA**TA
TTAATATAATTA**ATATTA**ATTTTTTCAATA**ATTTA**TTAA**ATTTA**TAT**ATTTA**TAATAAA
TTAAAT**ATTTA**A**TATATATATATATATATA**TATTAAACCGTTTTTAATAATTTTACATA
TAAATAA**AAAAAATTA**−*trnM*-**5, 484**

*Pandesma quenavadi*
*rrnS* - **3, 319** TT**TATATA**T**ATTA**TTTTTCAC**ATAGA****TTTTTTTTTTTTTTTTTTTTTTT**TAT**AT**
**TTA**AAT**ATTTA**A**TATATATATA**AATGT**ATATTA****AAAT****ATTTA****ATATAATTATTAAAT**
**ATTA****AATAATT**TCTCTTTTTTTTTTCTTTATAATATTCATAATAAAACCAAATTTGG**AA**
**ATTAAACAATTACAATTCTTAAAAATTACAAT**ATATTA**ATATAATTG**ATATTA**ATTCTT
TCAGTG**ATTTA**AAGTATTATTAAT**ATATTA**AT**ATTTA**TTTAAAT**ATTTA**A**TATATATAT**
**ATATATA**GTGAAATTTCTAAGAATTTTTTCATTCATGTTAGTTTTTAGACCATTTTTAA
TAATTTTTCATTAAATATAAAAAATATA−*trnM*-**3, 693**

*Polydesma boarmoides*
*rrnS*- **15, 387** TTTATATGTATTATTTTTCAC**ATAGA****TTTTTTTTTTTTTTTTTTTTTT**TAT**AT**
**TTA**AAT**ATTTA**ATATA**ATATTA**T**ATTTA**CT**ATATTA**AAT**ATTTA****ATTTA**AATTAATCAT
**TTAATT****TA**TATG**TATATATATATATATATA****ATTTA**TG**TATATATATATATATA****ATA**TTATT
**ATTATATATATACAT**ATAT**TAATAATTG**TATTAAACATTAAATAATTTCTTTTATTT
TTTCTTTATAATATTCTTAATAAATACAAATTTGGAAATTAAACAATTATAATTCTTAA
AAATTACAAT**ATATTA**AT**ATATTA**AT**ATTA**ATTATCTTAGTTAAGTTAATGAAT**TTTAA**
**ATATTA**TATTAT**ATTTA**AAT**ATTTA**AT**ATTTA**A**ATATTA**T**ATTTA**AATTTTTAATATAA
TATTTTAACTATTAT**ATTTA**AAT**ATTTA**ATATAATTA**ATTTA**T**ATTTA**AAT**ATTTA**ATA
**TAAT**GGTATAATTATG**TATATATATATATA**AATTTTT**TATATATATATATA**AAAT−
*trnM*-**46**

**Fig 9. Motifs, microsatellites and tandem repeats found in the A+T-rich region of *Episparis tortuosalis*, *Erebus macrops*, *Xanthodes albago*, *Pandesma quenavadi*, *Polydesma boarmoides*.** These are indicated by specific colours and highlights. Motifs (ATAGA) are shown in bold dark red. Poly-T stretches are shown in dark blue. Poly-A stretches are shown in green colour. Microsatellites (ATATTA) are shown in pink highlights. Microsatellites (ATTTA) are shown in red highlight. All tandem repeats are bold underlined. Microsatellite (TA)3, (TA)5, (TA)6, (TA)7, (TA)9 and (TA)10 are shown in turquoise highlights.

Phylogenetic analyses were performed based on the concatenated alignment of *13PCGs_AA* covering 225 Macro-heteroceran species from five superfamilies: Noctuoidea (191), Bombycoidea (15), Geometroidea (13), Lasiocampoidea (4), and Drepanoidea (2). We selected two Papilionoidea species (*Papilio polytes* and *Trogonoptera brookiana*) as the outgroup. The phylogenetic analyses generated almost identical topologies with a little divergence based on the identical dataset matrices. The clear variance between the Maximum likelihood and Bayesian Inference trees are the phylogenetic position of the subfamily Calpinae and superfamily Drepanoidea in addition to some clade nodes having poor support values (Figs 10a and 10b). Additional mitogenomes and other molecular evidence might be necessary to solve these issues. Both the ML and BI topological structures, the ML result with high support values was analogous to other studies,

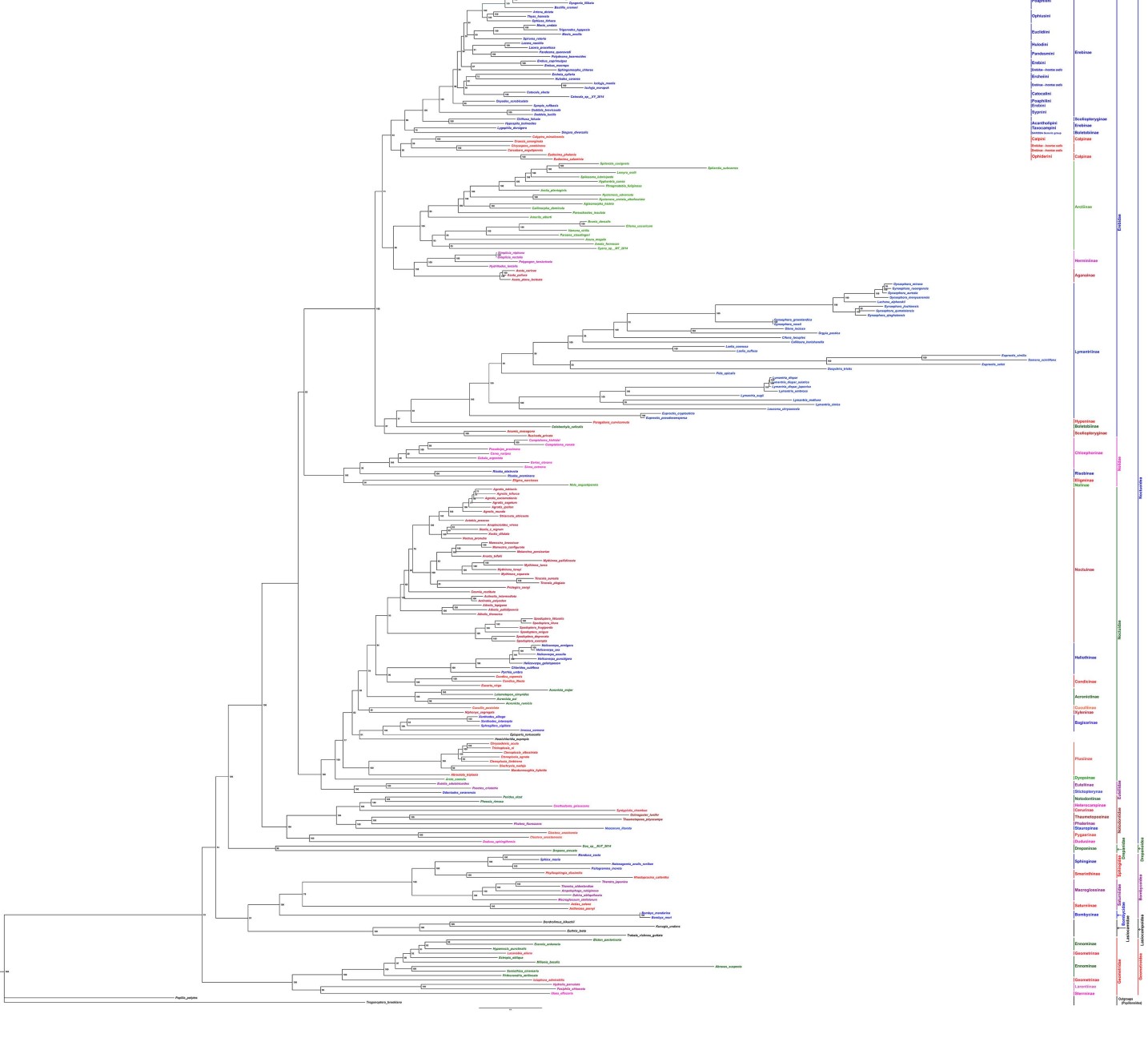

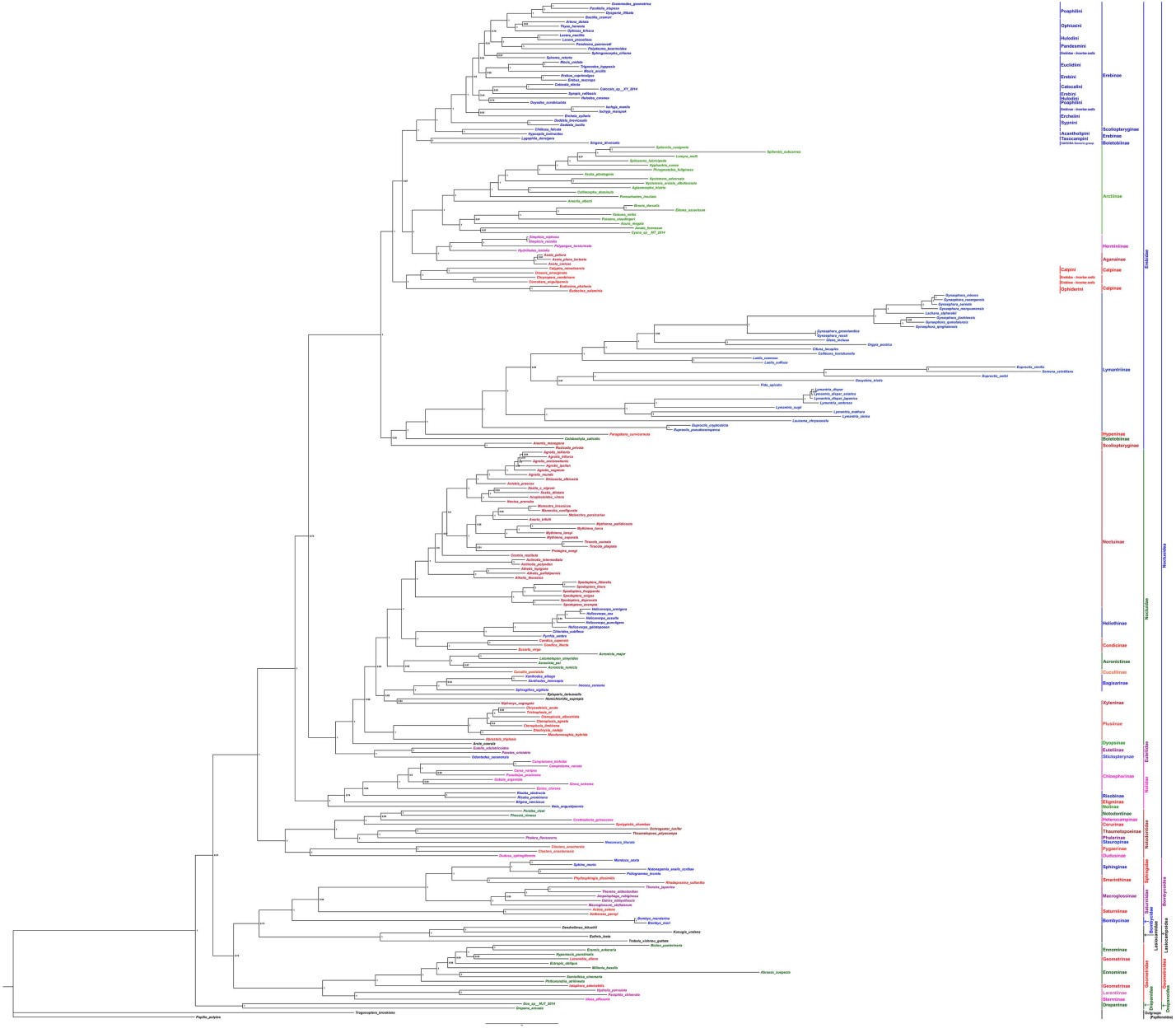

**Fig 10. a. Phylogenetic tree of Macroheterocera clade superfamilies using IQ-TREE.** The phylogenetic tree was reconstructed using 13 PCGs of the 225 species with maximum likelihood (ML) method (1000). The species *Papilo polytes* and *Trogonoptera brookiana* mitogenomes were used as outgroups. b. Phylogenetic tree of Macroheterocera clade superfamilies using MrBayes. The phylogenetic tree was reconstructed using 13 PCGs of the 225 species with Bayesian Inference. Posterior probability (1 = 100).

recommending that ML analysis might be more appropriate for discussing the phylogenetic relationships compared to the BI analysis [3,10,12,26,27,43,104].

van Nieukerken et al. [1] assigned Bombycoidea, Geometroidea, Lasiocampoidea, and Noctuoidea superfamilies to the Macroheterocera clade in the revised Lepidoptera system. According to earlier phylogenetic analyses based on various evidence, the Macroheterocera clade of Lepidoptera's phylogeny is still unstable [3–5]. Despite the limited taxon sampling, the monophyly of the five superfamilies was clearly validated in our studies using [1] new Lepidoptera family categorization system.

In the current study, ML analysis showed the following phylogenetic relationship ((Geometroidea)+ ((Lasiocampoidea+ Bombycoidea)+ (Drepanoidea)+ (Noctuoidea))))) (Fig 11a). Contrastingly, according to BI results the relationship between superfamilies was ((Drepanoidea+((Geometroidea+ (Lasiocampoidea+ Bombycoidea) + (Noctuoidea))))) (Fig 11b); this showed a novel relationship compared to previous analyses [4–9,12–15,27,43,105]. Specifically, we found that the superfamily Drepanoidea occupies a basal position within the Macroheterocera, rather than being part of the Bombycoid Complex (Bombycoidea, Lasiocampoidea, and Geometroidea). This led us to hypothesize that Drepanoidea represents

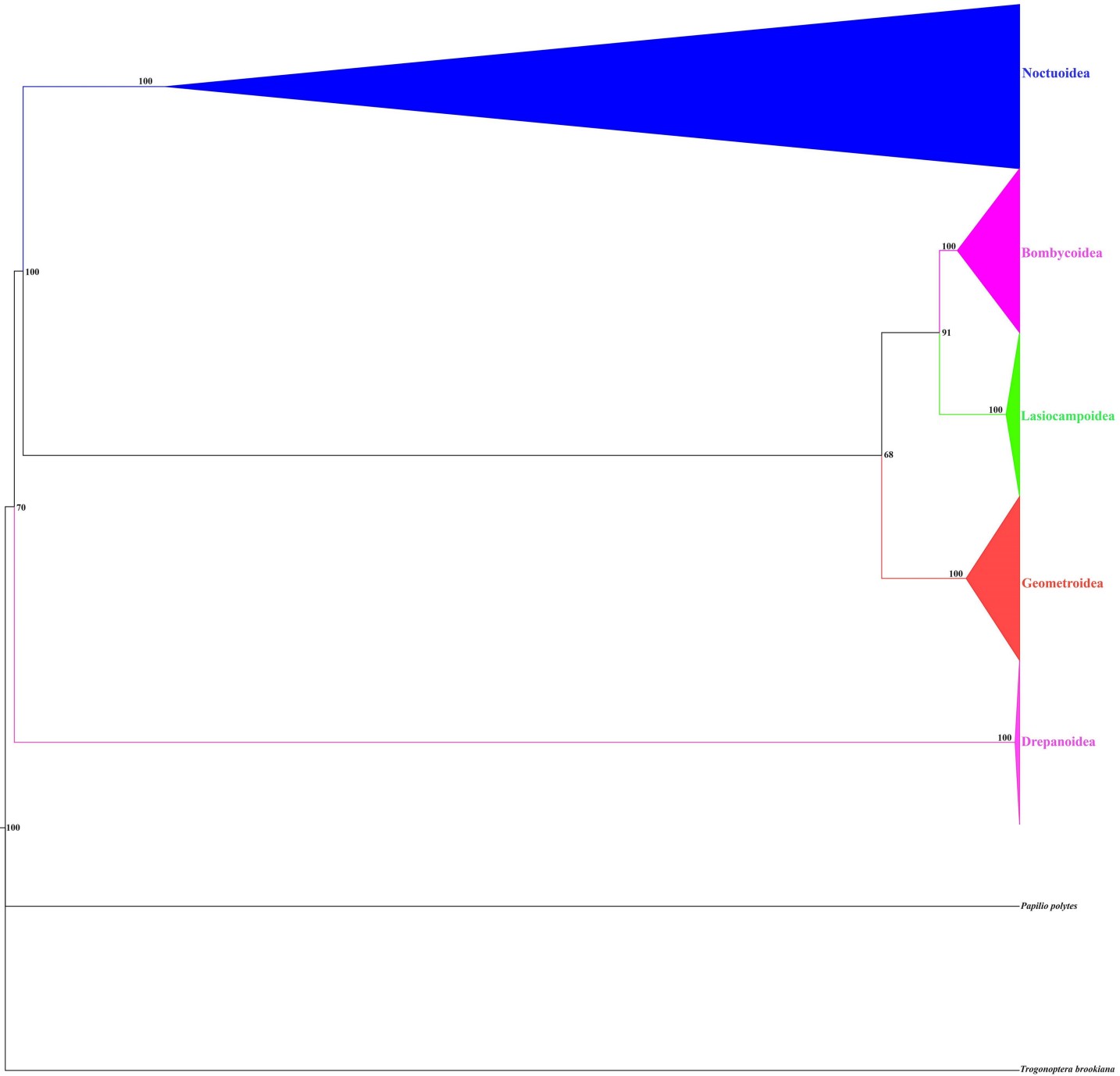

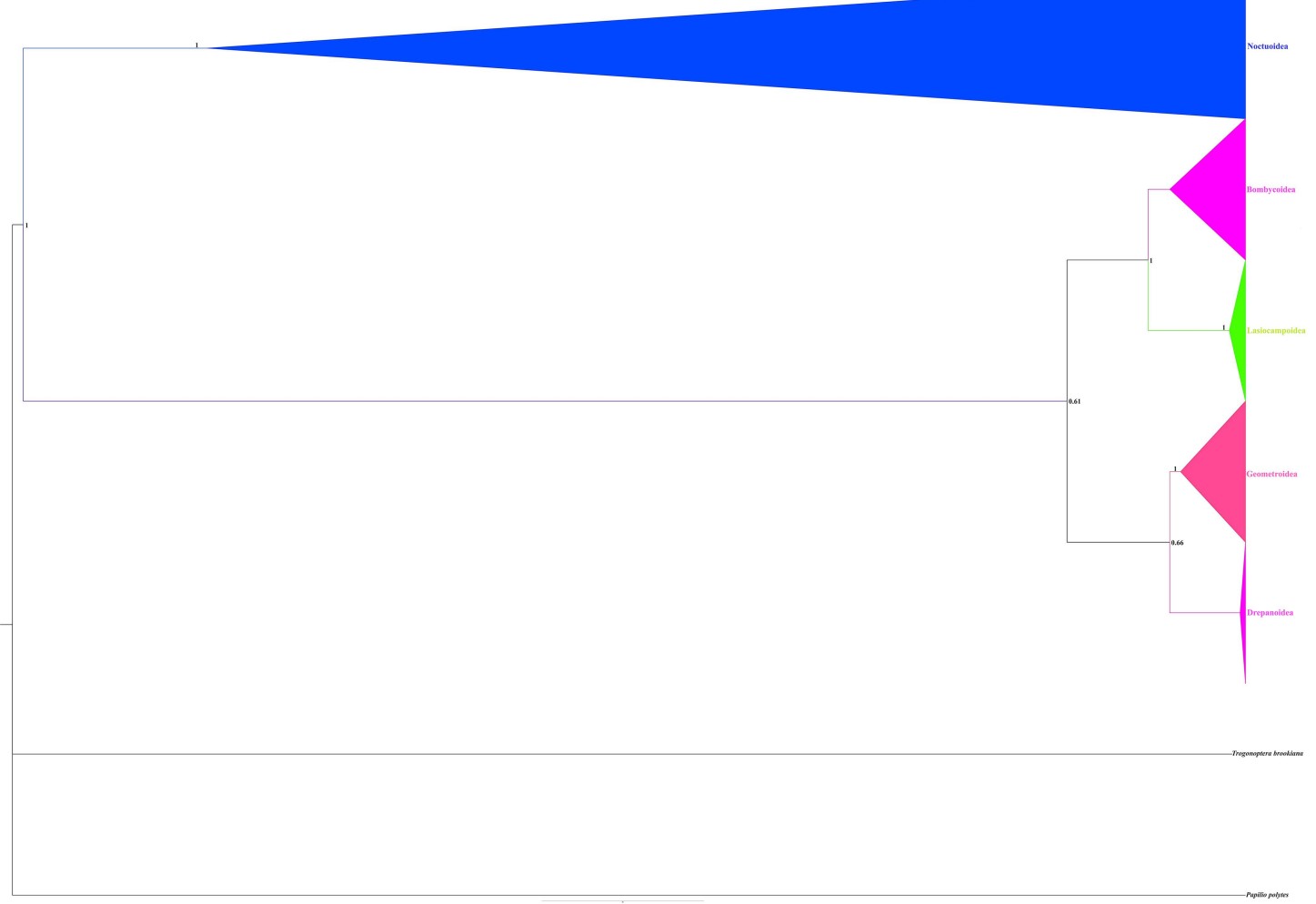

**Fig 11. a. Phylogenetic tree of the Macroheterocera clade based on maximum likelihood analysis. b. Phylogenetic tree of the Macroheterocera clade based on Bayesian Inference.**

the first divergent superfamily of the Macroheterocera clade and is sister to the remaining lepidopteran superfamilies in our current analysis.

Geometroidea and Noctuoidea are the two most diversified superfamilies of Macroheterocera [1,4]. Our analyses showed that Geometroidea was more closely related to Bombycoidea than Noctuoidea. Moreover, previous studies [8,13,14,87] supported that Geometroidea was closer to Bombycoidea than Noctuoidea based on the mitogenome analyses. The present analysis of the superfamilies Bombycoidea and Lasiocampoidea formed a sister relationship with a high support value (BP ≥ 91). Some studies classified Lasiocampoidea as a family under Bombycoidea [101,106,107]. Almost all previous investigations based on various data recovered the sister relationship between Bombycoidea and Lasiocampoidea [4,6–8,15]. Moreover, the superfamilies Bombycoidea and Lasiocampoidea clustered in a single clade with a high support value (BP ≥ 81).

The relationships within the superfamily Noctuoidea were analyzed based on the five newly sequenced species. Noctuoidea (42, 407) is the largest superfamily in the order Lepidoptera [1], with long-presented challenging evolutionary issues [32]. The Bootstrap proportion (BP ≥ 100) and Bayesian posterior probabilities (PP = 1) strongly supported

each family's monophyly. In our studies, the ML and BI both strongly supported (BP ≥ 91; PP = 1) Notodontidae, Euteliidae Noctuidae, Nolidae and Erebidae monophyly and the topology of phylogenetic relationships as (Notodontidae)+ (Erebidae+ (Nolidae+ Euteliidae+ Noctuidae)). Furthermore, our findings also revealed that, despite their strong support for a monophyletic assemblage, the connections between Erebidae and other four lineages, i.e., Notodontidae, Nolidae, Euteliidae, and Noctuidae are unclear. The sister group of Erebidae among the quadrifid noctuoids (Noctuidae, Nolidae, Euteliidae, and Erebidae) is still unresolved. According to our phylogenetic analysis, Erebidae is sister to the other quadrifids (Nolidae, Euteliidae, and Noctuidae). This finding is congruent with the previous studies which were reconstructed based on the mitogenomes [4,14,61,63,65,89,93,95,96,108–110]; however, it contrasted with others [5,9,30,32,42,67,77,105,111]. The BI analysis showed the family level relationships of superfamily Noctuoidea as (Notodontidae)+ ((Nolidae)+ (Erebidae+ (Euteliidae+ Noctuidae))). Based on the present analysis, Notodontidae is sister to the rest of Noctuoidea families.

A robust monophyletic group was the family Erebidae. A total of 101 species belonging to nine subfamilies clustered together within Erebidae, and their novel topology of the relationships as ((Scoliopteryginae+ Boletobiinae+ Hypeninae+ Lymantriinae)+ ((Aganainae + Herminiinae) + (Arctiinae)+ (Calpinae+ Erebinae))))) with well supported values. The subfamily relationships within Erebidae in our analysis differs from previous molecular inferences [14,30,32,36,38–40,112]. The analyses clearly support the monophyletic relationships of the nine subfamilies within the Erebidae and confirm that Herminiinae is sister to the Aganainae with strongly supported value. The close relationships between the subfamilies were consistent with earlier molecular analysis as well [40]. We also observed that the subfamily Lymantriinae is sister to the rest of Erebid subfamilies with strong support.

Regarding the deep-level phylogeny, the tree revealed in Fig 10a showed that the subfamily Erebinae for this study was composed of two well supported and two moderately supported clades (Toxocampini, Acantholipini, Sypnini, Euclidiini, Ophiusini, Poaphilini, Ercheiini, Pandesmini, Hulodini, Erebini, and Catocalini) with the relationships of (Toxocampini)+ (Acantholipini)+ (Sypnini+ (Erebini1 + Poaphilini1)+ (Ercheiini+ Catocalini)+ (Erebini)+ (Pandesmini+ Hulodini)+ (Euclidiini)+ (Ophiusini+ Poaphilini))))))))). The newly sequenced species *P. quenavadi* and *P. boarmoides* belong to the tribe Pandesmini clustered with *Lacera noctilio* and *L. procellosa* from tribe Hulodini with strongly supported values (BP ≥ 91; PP: 1). This result provides strong evidence that the tribe Pandesmini is sister to the tribe Hulodiini. Seven species were grouped together and formed a clade with strongly supported sister relationships containing two tribes Poaphilini + Ophiusini (Fig 10a). This clade is divided into two major assemblage tribes, Poaphilini (*Grammodes geometrica, Parallelia stuposa, Dysgonia illibata* and *Bastilla crameri*) and Ophiusini (*Artena dotata, Thyas honesta* and *Ophiusa tirhaca*) each of which has strong evidence for reciprocal monophyly (BP > 100; PP:1). This outcome is consistent with the previous findings [113]. The newly sequenced species *Erebus macrops* clustered with the same genus taxon *E. caprimulgus* belonging to the tribe Erebini.

The newly sequenced species *Episparis tortuosalis* belongs to the *EPISPARIS* generic group under the subfamily Pangraptinae in the family Erebidae based on morphological characteristics [114,115]. Albeit in the present analysis, *E. tortuosalis* nested within a Noctuidae clade (BP > 100; PP:1) and not in the family Erebidae. However, only one species was sequenced in the subfamily Pangraptinae to infer the phylogeny, and more data were needed to confirm the permanent taxonomic positions of *EPISPARIS* generic group. In Bagisarinae, the newly sequenced species *Xanthodes albago* aggregated with *Xanthodes intersepta* with high support values (BP ≥ 100; PP: 1) and clustered in a clade with (*Imosca coreana + Sphragifera sigillata*). The result showed that the *Xanthodes* is a sister taxon to *Imosca* and *Sphragifera* with Bagisarinae.

## Supporting information

**S1 Fig. Putative secondary structures of 22tRNAs found in mitochondrial genome of Episparis tortuosalis.** (TIF)

**S2 Fig. Putative secondary structures of 22tRNAs found in mitochondrial genome of Pandesma quenavadi.**
(TIF)

**S3 Fig. Putative secondary structures of 22tRNAs found in mitochondrial genome of Erebus macrops.**
(TIF)

**S4 Fig. Putative secondary structures of 22tRNAs found in mitochondrial genome of Polydesma boarmoides.**
(TIF)

**S5 Fig. Putative secondary structures of 22tRNAs found in mitochondrial genome of Xanthodes albago.**
(TIF)

**S1 Table. List of complete mitogenomes of Macroheterocera superfamilies.**
(DOCX)

**S2 Table. Nucleotide compositions and skewness in superfamily Noctuoidea mitogenomes.**
(DOCX)

**S3 Table. Relative synonymous codon usage of five newly sequenced mitogenomes.**
(DOCX)

## Acknowledgments

The authors are grateful to Dr J. Merline Shyla, Director, Entomology Research Institute, Loyola College, Chennai, India for providing the facilities.

## Author contributions

**Conceptualization:** Sivasankaran Kuppusamy, Muzafar Riyaz, Rauf Ahmad Shah.

**Data curation:** Sivasankaran Kuppusamy, Muzafar Riyaz, Rauf Ahmad Shah, Karuppasamy Paraman.

**Formal analysis:** Sivasankaran Kuppusamy, Rauf Ahmad Shah, Karuppasamy Paraman.

**Funding acquisition:** Sivasankaran Kuppusamy, Ignacimuthu Savarimuthu.

**Investigation:** Sivasankaran Kuppusamy.

**Methodology:** Sivasankaran Kuppusamy.

**Project administration:** Sivasankaran Kuppusamy.

**Resources:** Sivasankaran Kuppusamy.

**Software:** Sivasankaran Kuppusamy.

**Supervision:** Sivasankaran Kuppusamy.

**Validation:** Sivasankaran Kuppusamy.

**Visualization:** Sivasankaran Kuppusamy.

**Writing – original draft:** Sivasankaran Kuppusamy.

**Writing – review & editing:** Ignacimuthu Savarimuthu.

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
