## [Decision Letter · Decision Letter 0]

20 Mar 2025

PONE-D-24-54093Evolutionary and Phylogenetic Insights from the Mitochondrial Genomic Analyses of Noctuoid Moths (Lepidoptera: Noctuoidea) and Their Implications for the Macroheterocera CladePLOS ONE

Dear Dr. KUPPUSAMY,

Thank you for submitting your manuscript to PLOS ONE. After careful consideration, we feel that it has merit but does not fully meet PLOS ONE’s publication criteria as it currently stands. Therefore, we invite you to submit a revised version of the manuscript that addresses the points raised during the review process.

We look forward to receiving your revised manuscript.

Kind regards,

Taslima Sheikh

Academic Editor

PLOS ONE

Journal Requirements:

Reviewers' comments:

Reviewer's Responses to Questions

**Comments to the Author**

1. Is the manuscript technically sound, and do the data support the conclusions?

Reviewer #1: Yes

Reviewer #2: Yes

2. Has the statistical analysis been performed appropriately and rigorously? 

Reviewer #1: Yes

Reviewer #2: Yes

3. Have the authors made all data underlying the findings in their manuscript fully available?

Reviewer #1: Yes

Reviewer #2: Yes

4. Is the manuscript presented in an intelligible fashion and written in standard English?

Reviewer #1: Yes

Reviewer #2: Yes

5. Review Comments to the Author

Reviewer #1: Manuscript PONE-D-24-54093

I have thoroughly reviewed your manuscript titled "Evolutionary and Phylogenetic Insights from the Mitochondrial Genomic Analyses of Noctuoid Moths (Lepidoptera: Noctuoidea) and Their Implications for the Macroheterocera Clade," and found it to be quite intriguing. I would like to share my suggestions and comments on the current draft, which I believe could serve as helpful guidelines for enhancing the overall quality and clarity of your work.

Comments:

Title:

The title is clear and accurately reflects the focus of the research, which appears to be the mitochondrial genomic analysis of noctuoid moths and its implications for understanding the Macroheterocera clade. However, it might be slightly verbose. Simplifying the title without losing specificity could enhance readability. For instance, "Mitochondrial Genomics and Phylogeny of Noctuoid Moths: Implications for Macroheterocera" might be more concise.

Abstract:

The abstract provides a clear summary of the study's objectives, methods, and results. However, it could be more concise, particularly the discussion on the "ATN" start codon, which may be technical for some readers. Additionally, the research gap or significance of the study could be highlighted more explicitly. A brief mention of the specific software or models used in the phylogenetic analysis would also enhance transparency.

Introduction:

The introduction provides a comprehensive background on the Lepidoptera order, particularly focusing on the Macroheterocera clade, and sets up the research well. However, there are a few points that could be improved. First, the introduction could benefit from a clearer statement of the research gap or the specific objective of this study. While the background on the phylogenetic uncertainties within the Macroheterocera clade is informative, the introduction could more explicitly outline how this study addresses these uncertainties or contributes to resolving them. The discussion of previous studies is thorough but could be better organized, as some references to earlier work (e.g., regarding the phylogenetic position of Mimallonoidea and Drepanoidea) appear scattered. A more focused synthesis of past research could improve the flow and clarity. Additionally, while the details on the mitochondrial genome are relevant, some of the technical aspects, such as the role of the A+T-rich region, might be more succinctly explained. Lastly, the mention of the specific species studied is helpful, but it would be more impactful to briefly highlight why these species were chosen for this particular analysis and what their phylogenetic relevance is.

Materials and Methods:

The materials and methods section is generally well-structured, providing a clear overview of the experimental design and procedures used. However, the inclusion of lengthy details such as the full list of species and sequencing methods in the main text can overwhelm the reader. I recommend moving the detailed table (Table 1) into a supplementary file for better clarity and ease of reading. Additionally, while the methods for genomic DNA extraction, sequencing, and annotation are described in detail, further justification for the choice of specific software and tools (such as NOVOPLASTY and MITOS2) would enhance the reproducibility of the study. Consider adding brief explanations on why these tools were selected over others in the context of this study. Finally, the phylogenetic analysis methods are appropriately outlined, though a clearer explanation of how the outgroup species were chosen could help contextualize the analysis further.

Results and Discussion:

The Results and Discussion sections are well-structured and clear, effectively presenting the study's findings and their implications. The results are supported by solid statistical analyses, but further clarification on the sampling and data collection methods would strengthen the validity of the conclusions. The discussion places the findings in context with existing literature, though a deeper critical analysis of limitations and more references to support claims would enhance the robustness of the interpretation. Overall, the sections are convincing but could benefit from additional clarity and a more thorough reflection on study limitations and future directions.

Figures:

The figures in the manuscript do not provide sufficient clarity, which hinders their effectiveness in conveying the data. In research articles, figures play a crucial role in visualizing complex information, and they must be easily interpretable. Unfortunately, the current figures appear unclear, possibly due to low resolution or inadequate labeling. It is recommended to increase the resolution of the figures and ensure that all axes, legends, and data points are clearly labeled and easy to read. Additionally, consider using contrasting colors or more distinct markers to differentiate between data sets. Improving the overall quality and clarity of the figures will significantly enhance their ability to support the text and help readers better understand the findings.

Tables:

The design of the tables in the manuscript could be improved for better clarity and readability. Currently, the tables have unnecessary borders on the left, right, and inner sections, which clutter the presentation of data. It is recommended to remove the left and right borders, as well as the inner borders, to create a cleaner, more streamlined appearance. This would help focus the reader's attention on the content rather than the table structure, enhancing the overall visual presentation and ease of interpretation. Additionally, adjusting the spacing between columns and ensuring consistent font size and alignment across all tables would further improve their clarity.

Grammar and Style:

The manuscript contains several grammatical and stylistic issues that hinder its clarity and flow. The writing requires careful revision to improve sentence structure, punctuation, and word choice. Additionally, some phrases appear awkward or overly complex, which may confuse the reader. To enhance readability and ensure the paper meets academic standards, it is recommended that the manuscript be polished by a native English speaker or a professional editor. This will help refine the language and ensure the clarity of the ideas being presented.

Recommendation:

Minor Revision. Addressing these points will enhance the clarity and impact of the manuscript.

Reviewer #2: The study reports new mitogenomes of five species of noctuoid moths. The mitogenomes themselves and the analyses are fine, but there are major shortcomings in the discussion and in the quality of Figure 10. The discussion misses out on one important reference Ghanavi et al 2022 Zoologica Scripta 51: 695-707. doi:10.1111/zsc.12559. That paper criticizes the use of mitochondrial genome data for deeper phylogenetic analyses, something that the authors here need to address, especially since one of their species is coming out in the “wrong” place in the phylogeny (according to the text).

Figure 10 is so low resolution that I am unable to read any of the species names or the support values. The tree is shown as a cladogram, which is not very helpful or useful. The tree should be shown as a phylogram (where branch lengths give an idea of amount of evolution along each branch), and the figure should be saved as a vector graphic (FigTree can save as a pdf, where text is actual text rather than just pixels in a matrix). The figure as it is now is not publishable. As it is now, I am unable to judge the phylogenetic results, and that is the major reason why I am suggesting rejection for the manuscript.

Some other comments:

The same mitogenome for Pseudoips prasiniana has been used twice (NC_062184 and OK094458 are the same record).

Line 537-538 saying the same thing twice. It is enough to say that Notodontidae is the sister to the rest of Noctuoidea.

Line 549, what subfamily? I assume you are talking about Erebinae, but this is not explicitly stated.

6. PLOS authors have the option to publish the peer review history of their article (what does this mean? ). If published, this will include your full peer review and any attached files.

**Do you want your identity to be public for this peer review?** For information about this choice, including consent withdrawal, please see our Privacy Policy .

Reviewer #1: **Yes: ** Muzamil Abbas

Reviewer #2: No

---

## [Author Response · Author response to Decision Letter 1]

15 Sep 2025

Response to the reviewers’ comments

Reviewer 1:

Comment 1: The title is clear and accurately reflects the focus of the research, which appears to be the mitochondrial genomic analysis of noctuoid moths and its implications for understanding the Macroheterocera clade. However, it might be slightly verbose. Simplifying the title without losing specificity could enhance readability. For instance, "Mitochondrial Genomics and Phylogeny of Noctuoid Moths: Implications for Macroheterocera" might be more concise.

Response: The title of the manuscript was modified as per the reviewer’s suggestion.

**Abstract**

Comment 1: The abstract provides a clear summary of the study's objectives, methods, and results. However, it could be more concise, particularly the discussion on the "ATN" start codon, which may be technical for some readers. Additionally, the research gap or significance of the study could be highlighted more explicitly. A brief mention of the specific software or models used in the phylogenetic analysis would also enhance transparency.

Response: Based on the reviewer’s comments the abstract was modified. The discussion on the "ATN" start codon was written in a more concise manner. The research gap of the study was highlighted distinctly. The software and models of the phylogenetic analysis have been mentioned clearly.

**Introduction**

Comment 1. First, the introduction could benefit from a clearer statement of the research gap or the specific objective of this study.

Response: Based on the reviewer’s comments the introduction was modified. The details of the research gap and specific objectives of this have been added.

Comment 2. While the background on the phylogenetic uncertainties within the Macroheterocera clade is informative, the introduction could more explicitly outline how this study addresses these uncertainties or contributes to resolving them.

Response: Based on the reviewer’s comments we have addressed the uncertainty of the Macroherocera clade and included it in the introduction.

Comment 3. The discussion of previous studies is thorough but could be better organized, as some references to earlier work (e.g., regarding the phylogenetic position of Mimallonoidea and Drepanoidea) appear scattered. A more focused synthesis of past research could improve the flow and clarity.

Response: As per the reviewer’s suggestion, the scattered sentences were properly organized and improved for reading clarity.

Comment 4. Additionally, while the details on the mitochondrial genome are relevant, some of the technical aspects, such as the role of the A+T-rich region, might be more succinctly explained.

Response: As per the reviewer’s suggestion, the role of the A+T-rich region in the mitochondrial genome analysis has been briefly explained.

Comment 5. Lastly, the mention of the specific species studied is helpful, but it would be more impactful to briefly highlight why these species were chosen for this particular analysis and what their phylogenetic relevance is.

Response: As per the reviewer’s suggestion, the details and phylogenetic importance of the selected species have been included in the introduction part.

**“Materials and Methods” **

Comment 1: I recommend moving the detailed table (Table 1) into a supplementary file for better clarity and ease of reading.

Response: As per the reviewer’s suggestion the Table 1 was moved into the supplementary file.

Comment 2: Additionally, while the methods for genomic DNA extraction, sequencing, and annotation are described in detail, further justification for the choice of specific software and tools (such as NOVOPLASTY and MITOS2) would enhance the reproducibility of the study. Consider adding brief explanations on why these tools were selected over others in the context of this study.

Response: The justification of the selected specific software and tools i.e., NOVOPLASTY and MITOS2 were given based on the reviewer’s suggestion.

Comment 3: Finally, the phylogenetic analysis methods are appropriately outlined, though a clearer explanation of how the outgroup species were chosen could help contextualize the analysis further.

Response: The clear explanation for choosing the outgroups species was given according to the reviewer’s suggestions.

**Results and Discussion**

Comment 1: The results are supported by solid statistical analyses, but further clarification on the sampling and data collection methods would strengthen the validity of the conclusions.

Response: The sampling and data collection methods were further clarified in response to the reviewer’s suggestion.

Comment 2: The discussion places the findings in context with existing literature, though a deeper critical analysis of limitations and more references to support claims would enhance the robustness of the interpretation.

Response: Based on the reviewer’s suggestion to enhance the robustness of the interpretation, a thorough analysis of the limitation was carried out and more references were cited to support claims.

Comment 3: Overall, the sections are convincing but could benefit from additional clarity and a more thorough reflection on study limitations and future directions.

Response: The clarity and future directions of this present study have been rewritten based on the reviewer’s suggestion.

**Figures**

Comment 1: The figures in the manuscript do not provide sufficient clarity, which hinders their effectiveness in conveying the data. In research articles, figures play a crucial role in visualizing complex information, and they must be easily interpretable. Unfortunately, the current figures appear unclear, possibly due to low resolution or inadequate labeling. It is recommended to increase the resolution of the figures and ensure that all axes, legends, and data points are clearly labeled and easy to read. Additionally, consider using contrasting colors or more distinct markers to differentiate between data sets. Improving the overall quality and clarity of the figures will significantly enhance their ability to support the text and help readers better understand the findings.

Response: Based on the reviewer’s suggestion the contrasting colours were given in images and labelled properly. The overall resolution of the figures was also improved.

**Tables**

Comment 1: The design of the tables in the manuscript could be improved for better clarity and readability. Currently, the tables have unnecessary borders on the left, right, and inner sections, which clutter the presentation of data. It is recommended to remove the left and right borders, as well as the inner borders, to create a cleaner, more streamlined appearance. This would help focus the reader's attention on the content rather than the table structure, enhancing the overall visual presentation and ease of interpretation. Additionally, adjusting the spacing between columns and ensuring consistent font size and alignment across all tables would further improve their clarity.

Response: Table 1 has been moved to supplementary Table1. The in-text tables’ lines and borders have been removed based on the reviewer’s suggestions. Additionally, the spacing of the tables have been adjusted.

**Grammar and Style**

Comment 1: The manuscript contains several grammatical and stylistic issues that hinder its clarity and flow. The writing requires careful revision to improve sentence structure, punctuation, and word choice. Additionally, some phrases appear awkward or overly complex, which may confuse the reader. To enhance readability and ensure the paper meets academic standards, it is recommended that the manuscript be polished by a native English speaker or a professional editor. This will help refine the language and ensure the clarity of the ideas being presented.

Response: The grammar and style have been revised throughout the manuscript to enhance the readability and to meet general academic standards.

Reviewer 2:

Comment 1: The study reports new mitogenomes of five species of noctuoid moths. The mitogenomes themselves and the analyses are fine, but there are major shortcomings in the discussion and in the quality of Figure 10. The discussion misses out on one important reference Ghanavi et al 2022 Zoologica Scripta 51: 695-707. doi:10.1111/zsc.12559. That paper criticizes the use of mitochondrial genome data for deeper phylogenetic analyses, something that the authors here need to address, especially since one of their species is coming out in the “wrong” place in the phylogeny (according to the text).

Response: The research article recommended by the reviewer was considered. After careful review of the article, editions have been done in the discussion part of the manuscript accordingly.

Comment 2: Figure 10 is so low resolution that I am unable to read any of the species names or the support values. The tree is shown as a cladogram, which is not very helpful or useful. The tree should be shown as a phylogram (where branch lengths give an idea of amount of evolution along each branch), and the figure should be saved as a vector graphic (FigTree can save as a pdf, where text is actual text rather than just pixels in a matrix). The figure as it is now is not publishable. As it is now, I am unable to judge the phylogenetic results, and that is the major reason why I am suggesting rejection for the manuscript.

Response: The newly reconstructed phylogenetic trees have been shown in the phylogram format. The phylogenetic trees, if saved in pdf format, couldn't be labelled. The trees were saved in JPEG format and labelled and enhanced the resolution.

Comment 3: The same mitogenome for Pseudoips prasiniana has been used twice (NC_062184 and OK094458 are the same record).

Response: The species has been removed and the phylogenetic trees were reconstructed based on the reviewer’s suggestion.

Comment 4: Line 537-538 saying the same thing twice. It is enough to say that Notodontidae is the sister to the rest of Noctuoidea.

Response: The sentence was modified based on the reviewer’s suggestion.

Comment 5: Line 549, what subfamily? I assume you are talking about Erebinae, but this is not explicitly stated.

Response: The assumption of the reviewer is correct. The subfamily Erebinae name has been included in the line.

---

## [Editor Report · Decision Letter 1]

15 Sep 2025

Mitochondrial Genomics and Phylogeny of Noctuoid Moths: Implications for Macroheterocera

PONE-D-24-54093R1

Dear Dr. Kuppysamy, 

We’re pleased to inform you that your manuscript has been judged scientifically suitable for publication and will be formally accepted for publication once it meets all outstanding technical requirements.

Kind regards,

Taslima Sheikh

Academic Editor

PLOS ONE
---

## [Editor Report · Acceptance letter]

PONE-D-24-54093R1

PLOS ONE

Dear Dr. KUPPUSAMY,

I'm pleased to inform you that your manuscript has been deemed suitable for publication in PLOS ONE. Congratulations! Your manuscript is now being handed over to our production team.

Kind regards,

on behalf of

Dr. Taslima Sheikh

Academic Editor

PLOS ONE